# Everybody Needs Good Neighbours: An Unsupervised Locality-based Method for Bias Mitigation

**Xudong Han**[1,2]  **Timothy Baldwin**[1,2]  **Trevor Cohn**[1]
[1]The University of Melbourne
[2]Mohamed bin Zayed University of Artificial Intelligence (MBZUAI)
`xudongh1@student.unimelb.edu.au, {tbaldwin,t.cohn}@unimelb.edu.au`

## Abstract

Learning models from human behavioural data often leads to outputs that are biased with respect to user demographics, such as gender or race. This effect can be controlled by explicit mitigation methods, but this typically presupposes access to demographically-labelled training data. Such data is often not available, motivating the need for unsupervised debiasing methods. To this end, we propose a new meta-algorithm for debiasing representation learning models, which combines the notions of data locality and accuracy of model fit, such that a supervised debiasing method can optimise fairness between neighbourhoods of poorly vs. well modelled instances as identified by our method. Results over five datasets, spanning natural language processing and structured data classification tasks, show that our technique recovers proxy labels that correlate with unknown demographic data, and that our method outperforms all unsupervised baselines, while also achieving competitive performance with state-of-the-art supervised methods which are given access to demographic labels.

## 1 Introduction

It is well known that naively-trained models potentially make biased predictions even if demographic information (such as gender, age, or race) is not explicitly observed in training, leading to discrimination such as opportunity inequality (Hovy & Søgaard, 2015; Hardt et al., 2016). Although a range of fairness metrics (Hardt et al., 2016; Blodgett et al., 2016) and debiasing methods (Elazar & Goldberg, 2018; Wang et al., 2019; Ravfogel et al., 2020) have been proposed to measure and improve fairness in model predictions, they generally require access to protected attributes during training. However, protected labels are often not available (e.g., due to privacy or security concerns), motivating the need for unsupervised debiasing methods, i.e., debiasing without access to demographic variables. Previous unsupervised debiasing work has mainly focused on improving the worst-performing groups, which does not generalize well to ensuring performance parity across all protected groups (Hashimoto et al., 2018; Lahoti et al., 2020).

In Section 3, we propose a new meta-algorithm for debiasing representation learning models, named **U**nsupervised **L**ocality-based **P**roxy **L**abel assignment (ULPL). As shown in Figure 1, to minimize performance disparities, ULPL derives binary proxy labels based on model predictions, indicating poorly- vs. well-modelled instances. These proxy labels can then be combined with any supervised debiasing method to optimize fairness without access to actual protected labels. The method is based on the key observation that hidden representations are correlated with protected groups even if protected labels are not observed in model training, enabling the modelling of unobserved protected labels from hidden representations. We additionally introduce the notion of data locality to proxy label assignment, representing neighbourhoods of poorly- vs. well-modelled instances in a nearest-neighbour framework.

In Section 4, we compare the combination of ULPL with state-of-the-art supervised debiasing methods on five benchmark datasets, spanning natural language processing and structured data classification. Experimental results show that ULPL outperforms unsupervised and semi-supervised

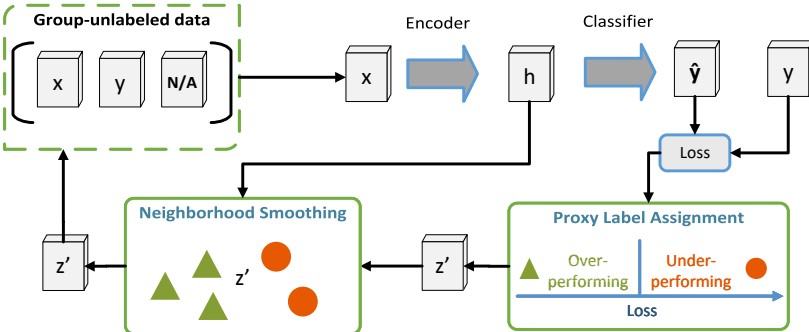

Figure 1: An overview of ULPL. Given a model trained to predict label y from $x$ by optimizing a particular loss, we derive binary proxy labels over- vs. under-performing within each target class based on training losses. These proxy labels are then smoothed according to the neighbourhood in latent space. Finally, the group-unlabeled data is augmented with $z'$, enabling the application of supervised bias mitigation methods.

baselines, while also achieving performance competitive with state-of-the-art supervised techniques which have access to protected attributes at training time.

In Section 5, we show that the proxy labels inferred by our method correlate with known demographic data, and that it is effective over multi-class intersectional groups and different notions of group-wise fairness. Moreover, we test our hypothesis of locality smoothing by studying the predictability of protected attributes and robustness to hyperparameters in finding neighbours.

## 2 RELATED WORK

**Representational fairness**   One line of work in the fairness literature is on *protected information leakage*, i.e., bias in the hidden representations. For example, it has been shown that protected information influences the geometry of the embedding space learned by models (Caliskan et al., 2017; May et al., 2019). Previous work has also shown that downstream models learn protected information such as authorship that is unintentionally encoded in hidden representations, even if the model does not have access to protected information during training (Li et al., 2018; Wang et al., 2019; Zhao et al., 2019; Han et al., 2021b). Rather than reduce leakage, in this work, we make use of leakage as a robust and reliable signal of unobserved protected labels and derive proxy information from biased hidden representations for bias mitigation.

**Empirical fairness**   Another line of work focuses on empirical fairness by measuring model performance disparities across protected groups, e.g., via demographic parity (Dwork et al., 2012), equalized odds and equal opportunity (Hardt et al., 2016), or predictive parity (Chouldechova, 2017). Based on aggregation across groups, empirical fairness notions can be further broken down into group-wise fairness, which measures relative dispersion across protected groups (Li et al., 2018; Ravfogel et al., 2020; Han et al., 2022a; Lum et al., 2022), and per-group fairness, which reflects extremum values of bias (Zafar et al., 2017; Feldman et al., 2015; Lahoti et al., 2020). We follow previous work (Ravfogel et al., 2020; Han et al., 2021b; Shen et al., 2022) in focusing primarily on improving group-wise equal opportunity fairness.

**Unsupervised bias mitigation**   Recent work has considered semi-supervised bias mitigation, such as debiasing with partially-labelled protected attributes (Han et al., 2021a), noised protected labels(Awasthi et al., 2020; Wang et al., 2021; Awasthi et al., 2021), or domain adaptation of protected attributes (Coston et al., 2019; Han et al., 2021a). However, these approaches are semi-supervised, as true protected labels are still required for optimizing fairness objectives.

Although Gupta et al. (2018) has proposed to use observed features as proxies for unobserved protected labels, the selection of proxy features is handcrafted and does not generalize to unstructured

inputs (e.g., text or images). Therefore, there is no guarantee of correlation between proxy labels and unobserved protected labels (Chen et al., 2019).

The most relevant line of work focuses on the notion of Max-Min fairness (Rawls, 2001), which aims to maximize the minimum performance across protected groups. Hashimoto et al. (2018) optimize worst-performing distributions without access to actual protected labels, but suffer from the risk of focusing on outliers, reducing the effectiveness of bias mitigation. Adversarially reweighted learning (ARL) (Lahoti et al., 2020) extends the idea by employing an additional adversary in training to prevent the optimization from focusing on noisy outliers, based on the notion of computational identifiability (Hébert-Johnson et al., 2017). However, adversarial training is notoriously non-convex, and there is no guarantee that the adversary will learn contiguous regions rather than identifying outliers. In contrast, our proposed neighbourhood smoothing method is memory-based, does not require adversarial training, and one can explicitly adjust the smoothness of neighbourhood search.

**Unsupervised fairness evaluation**    To access fairness without demographics, recent work (Kallus et al., 2020) has proposed to measure fairness w.r.t. auxiliary variables such as surname and geolocation in different datasets, which is a different research topic and beyond the scope of this paper. In this paper, we use protected labels for tuning and evaluation, and in practice, one can employ our unsupervised debiasing methods together with unsupervised fairness evaluation approaches to perform hyperparameter tuning for better fairness.

**Dataset cartography**    Training instances are also grouped based on predictability in the literature on dataset cartography, which is similar to the assignment of proxy labels in this paper. Swayamdipta et al. (2020) propose to visualize training instances according to variability and confidence, where a higher-confidence indicates the instance label can be predicted more easily. Le Bras et al. (2020) also group training instances by their predictability, measured by training simple linear discriminators. Such methods focus on improving in- and out-of-distribution performance without taking fairness into consideration. In comparison, our proposed method aims to mitigate bias by assigning proxy protected group labels to training instances based on their losses within a particular class.

## 3  METHODS

### 3.1  PROBLEM FORMULATION

Consider a dataset consisting of $n$ instances $\mathcal{D} = \{(\boldsymbol{x}_i, \mathrm{y}_i, \mathrm{z}_i)\}_{i=1}^n$, where $\boldsymbol{x}_i$ is an input vector to the classifier, $\mathrm{y}_i \in [1, \ldots, \mathrm{C}]$ represents target class label, and $\mathrm{z}_i \in [1, \ldots, \mathrm{G}]$ is the group label, such as gender. For unsupervised bias mitigation, protected labels are assumed to be unobserved at training and inference time. $n_{\mathrm{c,g}}$ denotes the number of instances in a subset with target label c and protected label g, i.e., $\mathcal{D}_{\mathrm{c,g}} = \{(\boldsymbol{x}_i, \mathrm{y}_i, \mathrm{z}_i) | \mathrm{y}_i = \mathrm{c}, \mathrm{z}_i = \mathrm{g}\}_{i=1}^n$. A vanilla model ($m = f \circ e$) consists of two connected parts: the encoder $e$ is trained to compute the hidden representation from an input, $\boldsymbol{h} = e(\boldsymbol{x})$, and the classifier makes prediction, $\hat{\mathrm{y}} = f(\boldsymbol{h})$.

Let $\mathcal{L}^{\mathrm{c,g}} = \frac{1}{n_{\mathrm{c,g}}} \sum_{(\boldsymbol{x}_i, \mathrm{y}_i, \mathrm{z}_i) \in \mathcal{D}_{\mathrm{c,g}}} \ell(m(\boldsymbol{x}_i), \mathrm{y}_i)$ be the average empirical risk for $\mathcal{D}_{\mathrm{c,g}}$, where $\ell$ is a loss function such as cross-entropy. Similarly, let $\mathcal{L}^{\mathrm{c}}$ denote the average for instances with target label c ($\mathcal{D}_{\mathrm{c}}$), and $\mathcal{L}$ denote the overall empirical risk.

**Fairness measurement**    We follow previous work in measuring group-wise performance disparities (Ravfogel et al., 2020; Roh et al., 2021; Shen et al., 2022). Specifically, for a particular utility metric $U$, e.g., the true positive rate, the results for each protected group are C-dimensional vectors, one dimension for each class. For the subset of instances $\mathcal{D}_{\mathrm{c,g}}$, we denote the corresponding evaluation results as $U_{\mathrm{c,g}}$. Let $U_{\mathrm{c}}$ denote the overall utilities of class c, then group-wise fairness is achieved if the utilities of all groups are identical, $U_{\mathrm{c,g}} = U_{\mathrm{c}}, \forall \mathrm{c, g} \Leftrightarrow |U_{\mathrm{c,g}} - U_{\mathrm{c}}| = 0, \forall \mathrm{c, g}$. In addition to the overall performance metric $U$, we denote the fairness metric $F$, as a measurement of group-wise utility disparities.

### 3.2  UNSUPERVISED LOCALITY-BASED PROXY LABEL ASSIGNMENT

**Proxy label assignment**    To mitigate bias, the first question is how to minimize disparities of non-differentiable metrics in model training. Previous work has shown that empirical risk-based

objectives can form a practical approximation of expected fairness, as measured by various metrics including AUC-ROC (Lahoti et al., 2020), positive rate (Roh et al., 2021), and true positive rate (Shen et al., 2022). Without loss of generality, we illustrate with the equal opportunity fairness. Note that our method generalizes to other fairness criteria, see Appendix D for detailss.

By replacing the utility metrics $U$ with empirical risks w.r.t. an appropriate loss function $\mathcal{L}$, the group-wise fairness metrics are reformulated as $\sum_{c=1}^{C} \sum_{g=1}^{G} |\mathcal{L}^{c,g} - \mathcal{L}^{c}| = 0$, which is an approximation of the desired fairness measurement $\sum_{c=1}^{C} \sum_{g=1}^{G} |U_{c,g} - U_{c}| = 0$. However, protected labels (z) are not observed in unsupervised debiasing settings, which raises the question: how can we optimize fairness objectives with unobserved protected labels?

Based on the fairness objective $\sum_{c=1}^{C} \sum_{g=1}^{G} |\mathcal{L}^{c,g} - \mathcal{L}^{c}| = 0$, we propose to focus on groups within each target class that are systematically poorly modelled. To this end, we binarize the numerous unobserved group labels into two types based on their training losses: $z_i' = \mathbf{1}_{\mathcal{L}_i > \mathcal{L}_i^{y_i}}$, where $z_i'$ denotes the augmented proxy group label. The two types of protected labels indicate that the loss of an instance is either greater than the mean (an 'under-represented' group) or $\leq$ the mean (an 'over-represented' group). Each instance can now be assigned with a binary proxy label and used with existing debiasing methods, resulting in augmented datasets $\mathcal{D}' = \{(\boldsymbol{x}_i, y_i, z_i')\}_{i=1}^{n}$.

**Neighbourhood smoothing**   Simply focusing on worse-performing instances can force the classifier to memorize noisy outliers (Arpit et al., 2017), reducing the effectiveness of bias mitigation. To address this problem, we find the neighbourhood that is likely to be from the same demographic based on the observation of protected information leakage (introduced in Section 2, and justified in Section 5.2), and smooth the proxy label of each instance based on its neighbours.

Specifically, we use the notion of data locality and adopt a $k$-Nearest-Neighbour classifier ($k$-NN) to smooth the proxy label. Given hidden representations $\{\boldsymbol{h}_1, \boldsymbol{h}_2, \ldots, \boldsymbol{h}_n\}$, where $\boldsymbol{h}_i = e(\boldsymbol{x}_i)$, and a query point $\boldsymbol{h}_j$, $k$-NN searches for the $k$ points $\{\boldsymbol{h}_{j_1}, \boldsymbol{h}_{j_2}, \ldots, \boldsymbol{h}_{j_k}\}, s.t.\ y_j = y_{j_k}$ is closest in distance to $\boldsymbol{h}_j$, and then makes predictions through majority vote among proxy labels $\{z_{j_1}', z_{j_2}', \ldots, z_{j_k}'\}$. Unlike the standard setting of $k$-NN, where the query point is excluded from consideration, we include the query instance. As a result, the smoothing process degrades to using naive proxy labels when $k = 1$, where the discriminator prediction is the original proxy label without smoothing. For $k > 1$, on the other hand, neighbourhood smoothing comes into effect.

Proxy label assignment and neighbour smoothing can be applied at different granularities, such as calculating the loss at steps vs. iterations; see Appendix C.1 for details.

### 3.3   THEORETICAL JUSTIFICATION

**Approximating Fairness Criteria**   In multi-class classification settings, the equal opportunity fairness is achieved if $\hat{y} \perp z|y, \forall y$, i.e., the true positive rates (TPR) of each target class are equal for all partitions of the dataset, where partitioning is based on z.

Using the definition of cross-entropy of the $i$-th instance, $-\sum_{c=1}^{C} \mathbb{1}_{\{y_i\}}(c) \log(p(\hat{y}_i = c))$ where $\hat{y}_i$ is a function of $\boldsymbol{x}_i$, the loss for the subset of instances with target label c can be simplified as:

$$\mathcal{L}^c = \frac{1}{n_c} \sum_{(\boldsymbol{x}_i, y_i, z_i) \in \mathcal{D}_c} -\log(p(\hat{y}_i = c)) = \frac{1}{n_c} \sum_{(\boldsymbol{x}_i, y_i, z_i) \in \mathcal{D}} -\log(p(\hat{y} = c|y_i = c)) \tag{1}$$

Notice that $\mathcal{L}^c$ is calculated on the subset $\mathcal{D}_c = \{(\boldsymbol{x}_i, y_i, z_i)|y_i = c\}_{i=1}^{n}$, making $\mathcal{L}^c$ an unbiased estimator of $-\log p(\hat{y} = c|y_i = c)$, which approximates $-\log$ TPR of the c-th class. As such, it can be seen that TPR can be empirically replaced by cross-entropy loss when measuring EO fairness.

**Fairness Lower Bound**   Consider the worst case in fairness, e.g. $p(\hat{y} = 1, y = 1|z = 1) \approx 0$ and $p(\hat{y} = 1, y = 1|z = 2) \approx 1$, where the TPR gap between the two groups is 1. Such unfairness in training is shown as the minimum training loss of instances in group 1 being larger than the maximum loss of instances in group 0. Taking the proxy label assignment into consideration, this example results in the strong correlation between the gold-class group label and proxy group labels, i.e., $p(z' = 1|z = 1) = p(z' = 0|z = 2) \approx 1$. Therefore, the correlation between proxy labels

and gold-class group labels is positively correlated with unfairness, and the optimization w.r.t. proxy labels increases the lower bound of fairness.

# 4 EXPERIMENTAL RESULTS

This section demonstrates the effectiveness of our proposed method through experiments against various competitive baselines and across five widely-used datasets. We report evaluation results of all models based on average values over five runs with different random seeds for each dataset.

## 4.1 EXPERIMENT SETUP

**Datasets** We consider the following benchmark datasets[1] from the fairness literature: (1) **Moji** (Blodgett et al., 2016; Elazar & Goldberg, 2018), sentiment analysis with protected attribute *race*; (2) **Bios** (De-Arteaga et al., 2019; Subramanian et al., 2021), biography classification with protected attributes *gender* and *economy*; (3) **TrustPilot** (Hovy et al., 2015), product rating prediction with protected attributes *age*, *gender*, and *country*; (4) **COMPAS** (Flores et al., 2016), recidivism prediction with protected attributes *gender* and *race*; and (5) **Adult** (Kohavi, 1996), income prediction with protected attributes *gender* and *race*.

To enable thorough comparison and explore correlation with unobserved protected attributes, for datasets with more than one protected attribute, we treat each protected attribute as a distinct task, e.g., **Bios**–gender, and **Bios**–economy are treated as two different tasks. As a result, there are ten different tasks in total.

**Baselines** We employ the following baselines: (1) Vanilla, which trains the classifier without explicit bias mitigation; (2) FairBatch (Roh et al., 2021), which adjusts the resampling probabilities of each protected group for each minibatch to minimize loss disparities; (3) $\text{GD}_{\text{CLA}}$ (Shen et al., 2022), which adjusts the weights of each protected group to minimize loss disparities; (4) $\text{GD}_{\text{GLB}}$ (Shen et al., 2022), which is a variant of $\text{GD}_{\text{CLA}}$ that additionally minimizes loss differences across target classes; (5) Adv (Li et al., 2018), which trains the adversary to identify protected information from hidden representations, and removes protected information through unlearning adversaries; (6) SemiAdv (Han et al., 2021a), which trains the adversary with partially-observed protected labels; and (7) ARL (Lahoti et al., 2020), which employs an adversary to assign larger weights to computationally-identifiable underrepresented instances. Besides the Vanilla model, methods (2)-(5) are supervised debiasing baselines, SemiAdv is a semi-supervised debiasing baseline method, and ARL is the baseline for unsupervised bias mitigation.

In terms of our proposed method, we examine its effectiveness in combination with several supervised debiasing methods, $\text{GD}_{\text{CLA}}$, $\text{GD}_{\text{GLB}}$, and Adv, denoted ULPL+$\text{GD}_{\text{CLA}}$, ULPL+$\text{GD}_{\text{GLB}}$, and ULPL+Adv, respectively. To be clear, the supervision in each case is based on the proxy labels $z_i'$ learned in an unsupervised manner by ULPL.

**Evaluation Metrics** This paper is generalizable to different metrics by varying the objectives of the debiasing methods. For illustration purposes, we follow Ravfogel et al. (2020); Shen et al. (2022) and Han et al. (2021a) in measuring the overall accuracy and equal opportunity fairness, which measures true positive rate (TPR) disparities across groups. Consistent with Section 3.1, we measure the sum of TPR **gap** across subgroups to capture absolute disparities. We focus on less fair classes by using root mean square aggregation for class-wise aggregation. Overall, the fairness metric is $F = 1 - \sqrt{\frac{1}{\text{C}} \sum_{c=1}^{\text{C}} \left( \frac{1}{\text{G}} \sum_{g=1}^{\text{G}} |\text{TPR}_{c,g} - \text{TPR}_c| \right)^2}$. For both metrics, larger is better.

**Model comparison** Previous work has shown that debiasing methods suffer from performance–fairness trade-offs in bias mitigation (Shen et al., 2022). Most debiasing methods involve a trade-off hyperparameter to control the extent to which the model sacrifices performance for fairness, such as $\lambda_{\text{GD}_{\text{CLA}}}$, the strength of additional regularization objectives of $\text{GD}_{\text{CLA}}$ and our proposed method ULPL+$\text{GD}_{\text{CLA}}$. As shown in Figure 2a, given the performance–fairness trade-offs, selecting the

---

[1]Key characteristics of the datasets, including dataset statistics, are provided in Appendix A.

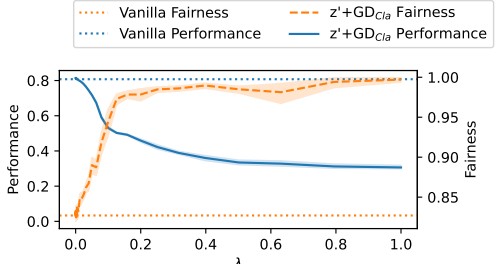 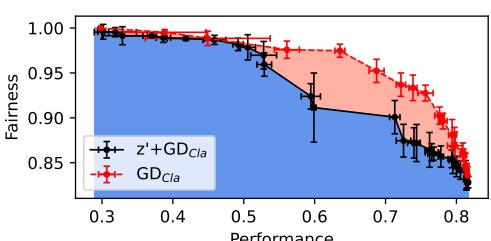

(a) Tuning ULPL+GD$_{\text{CLA}}$ trade-off hyperparameter. Shaded areas = 95% CI estimated over 5 runs.

(b) Performance–fairness curve of GD$_{\text{CLA}}$ (red dashed line), and ULPL+GD$_{\text{CLA}}$ (black solid line).

Figure 2: ULPL+GD$_{\text{CLA}}$ degrades to Vanilla performance (dotted lines) when setting $\lambda = 0$. As increase the weight of the fairness objective, fairness (orange dashed line) improves at the cost of performance (blue solid line). Figure 2b focuses on the Pareto frontier, and presents AUC-PFC as shaded area over the **Bios**-gender dataset.

| Dataset | Moji | Bios | | TrustPilot | | | Adult | | COMPAS | |
|---|---|---|---|---|---|---|---|---|---|---|
| Attribute | $R$ | $G$ | $E$ | $G$ | $A$ | $C$ | $G$ | $R$ | $G$ | $R$ |
| Vanilla | 0.172 | 0.471 | 0.490 | 0.130 | 0.128 | 0.125 | 0.092 | 0.082 | 0.111 | 0.101 |
| GD$_{\text{CLA}}$ | **0.249** | **0.498** | **0.507** | **0.133** | 0.131 | **0.132** | 0.095 | 0.084 | **0.124** | 0.100 |
| GD$_{\text{GLB}}$ | 0.230 | 0.480 | 0.495 | 0.132 | **0.131** | 0.131 | **0.096** | **0.085** | 0.124 | 0.102 |
| FairBatch | 0.245 | 0.482 | 0.495 | 0.130 | 0.131 | 0.130 | 0.094 | 0.083 | 0.119 | **0.106** |
| Adv | 0.247 | 0.484 | 0.494 | 0.132 | 0.132 | 0.129 | 0.094 | 0.084 | 0.120 | 0.102 |
| SemiAdv | 0.247 | 0.483 | 0.495 | 0.131 | 0.132 | 0.131 | 0.094 | 0.082 | 0.121 | 0.097 |
| ARL | 0.193 | 0.461 | 0.484 | 0.129 | 0.132 | 0.129 | 0.097 | 0.086 | 0.123 | 0.096 |
| ULPL+GD$_{\text{CLA}}$ | 0.209 | 0.485 | 0.503 | 0.133 | 0.132 | 0.131 | 0.093 | 0.082 | 0.126 | 0.105 |
| ULPL+GD$_{\text{GLB}}$ | 0.190 | 0.474 | 0.495 | 0.133 | 0.132 | 0.131 | 0.094 | 0.085 | 0.127 | 0.105 |
| ULPL+Adv | 0.185 | 0.472 | 0.493 | 0.132 | 0.131 | 0.131 | 0.094 | 0.084 | 0.125 | 0.101 |

Table 1: AUC-PFC on five datasets w.r.t. different protected attributes: $G$: gender; $E$: economy; $A$: age; $C$: country; $R$:race. Debiasing methods are introduced in Section 4. AUC-PFC scores are calculated based on trade-off curves averaged over 5 repeated runs with different random seeds.

best performance degrades to vanilla training and choosing the best fairness results in random predictions. As such, performance and fairness must be considered simultaneously in model comparisons, such as early stopping and hyperparameter tuning. We use protected attributes for early stopping over a validation set and report results on the test set. In practice, model selection should be made in a domain-specific manner, where the best method varies. To make quantitative comparisons based on the performance–fairness trade-offs, we follow Han et al. (2023) in reporting the area under the performance–fairness trade-off curves (AUC-PFC) of each method. As shown in Figure 2b, the performance–fairness trade-off curve (PFC) of a particular method consists of a Pareto frontier, which represents the best results that can be achieved in different scenarios, and the area under the curve based on PFC (AUC-PFC) reflects the overall goodness of a method. In particular, the AUC-PFC score of GD$_{\text{CLA}}$ (red) and ULPL+GD$_{\text{CLA}}$ (blue) are $0.498$ and $0.485$, respectively, and their difference (i.e., area between the two curves) is $0.013$. See Appendix B for further details.

## 4.2 Can we mitigate bias without access to protected attributes?

Table 1 compares our proposed ULPL based methods against baselines.

**Vanilla and supervised debiasing baselines:** Compared with the Vanilla model, supervised debiasing baselines (GD$_{\text{CLA}}$, GD$_{\text{GLB}}$, FairBatch, and Adv) substantially improve fairness with relatively little performance cost, resulting in larger AUC-PFC scores. Among the four supervised de-

| Dataset | Moji | Bios | | TrustPilot | | | Adult | | COMPAS | |
|---|---|---|---|---|---|---|---|---|---|---|
| **Attribute** | $R$ | $G$ | $E$ | $G$ | $A$ | $C$ | $G$ | $R$ | $G$ | $R$ |
| $F$ | 0.636 | 0.837 | 0.915 | 0.963 | 0.971 | 0.960 | 0.951 | 0.751 | 0.894 | 0.672 |
| $r_{z'}$ | 0.996 | 0.656 | 0.214 | 0.056 | 0.045 | 0.034 | 0.600 | 0.691 | 0.983 | 0.855 |

Table 2: Fairness ($F$) and Pearson's $r_{z'}$ between proxy labels and signed gaps for Vanilla.

biasing methods, $GD_{CLA}$ outperforms other methods, which is consistent with previous work (Shen et al., 2022).

Similar to the baseline debiasing methods, debiasing w.r.t. proxy labels (ULPL+∗) also improves fairness over Vanilla, and achieves results that are competitive with supervised debiasing methods.

**Semi-supervised debiasing baselines:** We examine the effectiveness of SemiAdv by removing $50\%$ of the protected labels, i.e., the adversary is trained over a subset of training instances. Observe that SemiAdv achieves almost identical results to Adv, consistent with Han et al. (2021a). Although SemiAdv uses protected labels in training, it is substantially outperformed by the proposed unsupervised method ULPL+$GD_{CLA}$.

**Unsupervised debiasing baselines:** ARL (Lahoti et al., 2020) is also an unsupervised debiasing method that trains an adversary to predict the weights of each instance such that the weighted empirical risk is maximized. The training objective of ARL does not match with the definition of group-wise fairness, and as such, it results in lower AUC-PFC scores than our proposed methods, that explicitly optimize for performance parity across protected groups.

In terms of excluding outliers, the adversary of ARL is intended to predict smooth weights of instances from $(\boldsymbol{x}, \mathrm{y})$, such that the trained model focuses more on worse-performing instances and is discouraged from memorizing noisy outliers. However, our results show that our ULPL method is more robust and effective in implementing these notions.

**Different ULPL methods:** Among the proxy label-based methods, ULPL+$GD_{CLA}$ consistently outperforms other methods. ULPL+$GD_{CLA}$ calculates the loss differences separately before aggregation, which eliminates the influence of group size in debiasing and treats each group and class equally in optimization, which is better aligned with the evaluation metric.

Adv is a popular method for achieving representational fairness, and differs from ULPL+$GD_{CLA}$ and ULPL+$GD_{GLB}$ in that it directly optimizes for empirical fairness. Removing protected information from hidden representations requires accurate perceptions of the global geometry of particular protected groups. However, proxy labels are based on local loss differences within each class, meaning that the same proxy label in different classes may conflate different protected groups. As such, the combination of ULPL with Adv is less effective than the two other combinations.[2]

## 5 ANALYSIS

### 5.1 PROXY LABEL ASSIGNMENT

We first investigate if the ULPL labels are meaningful through the lens of Pearson's correlation ($r_{z'}$) between proxy labels and signed performance gaps. Given that $F$ is optimized if $\sum_{c,g} |\mathrm{TPR}_{c,g} - \mathrm{TPR}_c| = 0$, an instance $(\boldsymbol{x}_i, \mathrm{y}_i)$ should be assigned $\mathrm{z}'_i = 0$ if $\mathrm{TPR}_{\mathrm{y}_i, \mathrm{z}_i} - \mathrm{TPR}_{\mathrm{z}_i} < 0$, i.e., the unobserved group z is under-performing in class $\mathrm{y}_i$, and $\mathrm{z}'_i = 1$ otherwise. We calculate $r_{z'}$ between $P(\mathrm{z}' = 0 | \mathrm{y}, \mathrm{z})$ and $\mathrm{TPR}_{\mathrm{y},\mathrm{z}} - \mathrm{TPR}_{\mathrm{z}}$, and presents the results of Vanilla over each dataset in Table 2.

It can be seen that there exists a strong correlation $r_{z'}$ for all datasets other than **TrustPilot**, indicating that the unsupervised proxy label recovers demographic data and provides a strong signal for bias mitigation. We also observe that better fairness results in smaller $r_{z'}$, for example, for **TrustPilot** and **Bios**-$E$, which is not surprising as the gaps ($|\mathrm{TPR}_{c,g} - \mathrm{TPR}_c|$) are close to 0.

---

[2]See Appendix E.6 for further discussion.

| | Dataset | Moji | Bios | | TrustPilot | | | Adult | | COMPAS | |
|---|---|---|---|---|---|---|---|---|---|---|---|
| | Attribute | R | G | E | G | A | C | G | R | G | R |
| (a) | Leakage | 83 | 79 | 84 | 40 | 52 | 26 | 80 | 25 | 83 | 43 |
| (b) | ULPL+GD$_{\mathrm{CLA}}$ | 9 | 1 | 1 | 1 | 1 | 1 | 13 | 13 | 3 | 13 |
| | ULPL+GD$_{\mathrm{GLB}}$ | 7 | 1 | 1 | 1 | 1 | 1 | 13 | 7 | 9 | 13 |
| | ULPL+Adv | 3 | 1 | 1 | 1 | 1 | 1 | 9 | 13 | 3 | 5 |

Table 3: **(a)** Leakage (%) of protected attributes. **(b)** Best $k$ assignments of each method.

## 5.2 EFFECTIVENESS OF THE NEIGHBOURHOOD-SMOOTHING

In smoothing ULPL labels, we hypothesise that an instance's neighbours are likely from the same protected group. Except for instance itself in smoothing, the remaining nearest neighbours are essentially the results of a standard $k$-nearest-neighbour (KNN) model. Therefore, we perform analysis based on standard KNN models and investigate if the remaining nearest neighbours are helpful for label smoothing, i.e., are they from the same protected group as the target instance.

**Protected information predictability:** Proxy label smoothing is based on the hypothesis that there is a strong correlation between the hidden representations and the protected labels, even if protected labels are not observed during training. To test this hypothesis, we employ 1-NN for protected label prediction based on Vanilla hidden representations within each target class, using leave-one-out cross-validation over each batch to evaluate the predictability of protected attributes.

Table 3a presents the results (macro F1 score) for each protected attribute, from which we can see a strong correlation between unobserved protected labels and hidden representations.

Furthermore, in Appendix E.2, we show that although debiasing methods successfully reduce performance disparities in downstream tasks, leakage of protected attributes in debiased hidden representations is still high, consistent with previous work (Han et al., 2021b).

**Sensitivity to $k$-NN hyperparameters:** Estimations of protected labels can be affected by $k$-NN's hyperparameters, including: (1) $p$, the norm of distance; (2) whether nearest-neighbours must match the target class versus all classes; and (3) $k$, the number of nearest neighbours. We explore the sensitivity to these hyperparameters for the **Moji** dataset in Figure 3.

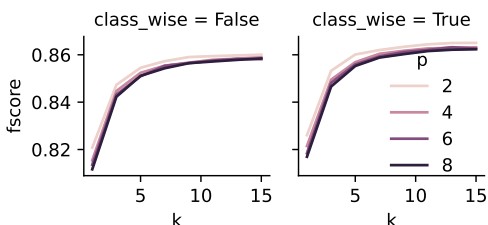

Figure 3: Hyperparameter sensitivity analysis over the **Moji** dataset.

First, we can see that $k$-NN is highly robust to varying values of $p$. In terms of whether label smoothing should be class-specific or inspecific, there is a slight empirical advantage to performing it in a class-specific manner.

In terms of $k$, for **Moji**, higher values result in better estimations of protected labels, although there is a clear plateau. If we explore this effect over the other datasets in terms of the $k$ value that achieves the highest AUC–PFC score, as can be seen in Table 3b, there is no clear trend, with neighbourhood-smoothing ($k > 1$) improving results for **Moji**, **Adult**, and **COMPAS** and the best results being achieved for values from 3 to 13, whereas for **Bios** and **TrustPilot**, no neighbourhood smoothing ($k = 1$) performs best. Although the optimal value of $k$ varies greatly across datasets and debiasing methods, it is possible to perform fine-grained tuning to reduce computational cost. In Appendix C.4, we discuss situations where label smoothing succeeds or fails, and an effective tuning strategy for the value of $k$.

## 5.3 DEBIASING FOR INTERSECTIONAL GROUPS

| Dataset | Moji | Adult | | | COMPAS | | |
|---|---|---|---|---|---|---|---|
| Attribute | $R$ | $G$ | $R$ | $G{\times}R$ | $G$ | $R$ | $G{\times}R$ |
| Vanilla | 0.173 | 0.087 | 0.076 | 0.073 | 0.109 | 0.097 | 0.094 |
| GD$_{\text{CLA}}$ | 0.253 | 0.091 | 0.081 | 0.079 | 0.121 | 0.094 | 0.085 |
| GD$_{\text{GLB}}$ | 0.233 | 0.091 | 0.081 | 0.077 | 0.121 | 0.098 | 0.094 |
| Adv | **0.254** | 0.091 | 0.077 | 0.074 | 0.117 | 0.094 | 0.945 |
| ARL | 0.197 | **0.092** | 0.081 | 0.078 | 0.120 | 0.092 | 0.092 |
| ULPL+GD$_{\text{CLA}}$ | 0.214 | 0.090 | 0.082 | **0.080** | 0.123 | **0.101** | 0.095 |
| ULPL+GD$_{\text{GLB}}$ | 0.192 | 0.090 | 0.081 | 0.078 | **0.124** | 0.100 | **0.097** |
| ULPL+Adv | 0.186 | 0.091 | **0.083** | **0.080** | 0.122 | 0.099 | 0.092 |

Table 5: AUC-PFC based on demographic parity fairness.

We also investigate the robustness of binary proxy labels to non-binary intersectional groups (i.e. the cross product of values across different protected attributes).

Table 4 presents debiasing results for intersectional groups over those datasets that are labelled with more than one protected attribute. Compared to the single protected attribute results, the AUC-PFC scores of Vanilla are consistently smaller, indicating greater bias across inter-

| Dataset | Bios | TrustPilot | Adult | COMPAS |
|---|---|---|---|---|
| Attribute | $G{\times}E$ | $G{\times}A{\times}C$ | $G{\times}R$ | $G{\times}R$ |
| Vanilla | 0.455 | 0.115 | 0.072 | 0.098 |
| GD$_{\text{CLA}}$ | **0.487** | **0.126** | 0.072 | 0.091 |
| GD$_{\text{GLB}}$ | 0.453 | 0.122 | 0.075 | 0.100 |
| Adv | 0.456 | 0.121 | 0.072 | 0.098 |
| ARL | 0.426 | 0.120 | 0.075 | 0.097 |
| ULPL+GD$_{\text{CLA}}$ | 0.470 | 0.105 | 0.076 | 0.098 |
| ULPL+GD$_{\text{GLB}}$ | 0.459 | 0.101 | **0.079** | **0.101** |
| ULPL+Adv | 0.456 | 0.122 | 0.071 | 0.099 |

Table 4: AUC-PFC w.r.t. intersectional groups.

sectional groups, consistent with the findings of Subramanian et al. (2021). For ULPL models, on the other hand, the results are competitive with supervised debiasing methods, consistent with the single attribute setting.

## 5.4 Other Fairness Metrics: Demographic Parity

Finally, we investigate the robustness of ULPL to other notions of fairness. For illustration purposes, we focus on demographic parity fairness (DP) (Blodgett et al., 2016), which requires model predictions to be independent of protected attributes. Again, we aggregate accuracy and DP fairness trade-offs as AUC-PFC scores. Since DP is sensitive to class imbalance and there is no standard way of generalizing DP to multi-class classification tasks, we only conduct experiments over binary classification tasks, namely **Moji**, **Adult**, and **COMPAS**.

Table 5 shows the results w.r.t. demographic parity fairness. The overall trend is similar to our original results for equal opportunity fairness, indicating that ULPL is robust to different fairness metrics when combined with a range of debasing methods.

## 6 Conclusion

Much of previous work in the fairness literature has the critical limitation that it assumes access to training instances labelled with protected attributes. To remove this restriction, we present a novel way of deriving proxy labels, enabling the adaptation of existing methods to unsupervised bias mitigation. We conducted experiments over five widely-used NLP and ML benchmark datasets, and showed that, when combined with different debiasing strategies, our proposed method consistently outperforms naively-trained models and unsupervised debiasing baselines, achieving results which are competitive with supervised debiasing methods. Furthermore, we showed our proposed method to be generalizable to multi-class intersectional groups and different notions of fairness.

ACKNOWLEDGEMENTS

We thank the anonymous reviewers for their helpful feedback and suggestions. This work was funded by the Australian Research Council, Discovery grant DP200102519. This research was undertaken using the LIEF HPC-GPGPU Facility hosted at the University of Melbourne. This Facility was established with the assistance of LIEF Grant LE170100200.

ETHICS STATEMENT

This work focuses on learning fair models without observation of protected labels at training time. Demographics are assumed to be available only for evaluation purposes, and not used for model training or inference. We only use attributes that the user has self-identified in our experiments. All data and models in this study are publicly available and used under strict ethical guidelines.

REPRODUCIBILITY STATEMENT

All baseline experiments are conducted with the *FairLib* library (Han et al., 2022b). Source code is available at `https://github.com/HanXudong/An_Unsupervised_ Locality-based_Method_for_Bias_Mitigation` Appendix A reports relevant statistics, details of train/test/dev splits, etc. for the five benchmark datasets that are used in this paper. Appendix B reports implementation details of evaluation metrics and corresponding aggregation approaches. Appendix C presents details of computing infrastructure used in our experiments, computational budget, hyperparameter search, etc. Appendix F reports PFC and full dis-aggregated results, i.e., mean $\pm$ std over 5 random runs with different random seeds.

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

# A DATASETS AND PRE-PROCESSING

## A.1 MOJI

Following previous studies (Ravfogel et al., 2020; Han et al., 2021b), the original training dataset is balanced with respect to both sentiment and ethnicity but skewed in terms of sentiment–ethnicity combinations (40% happy-AAE, 10% happy-SAE, 10% sad-AAE, and 40% sad-SAE, respectively). The dev and test sets are balanced in terms of sentiment–ethnicity combinations. The dataset contains 100K/8K/8K train/dev/test instances.

When varying training set distributions, we keep the 8k test instances unchanged. We use DeepMoji (Caliskan et al., 2017) to obtain twitter representations, where DeepMoji is a model pretrained over 1.2 billion English tweets and DeepMoji is fixed during model training.

## A.2 BIOS

**Bios** experiments are based on a biography classification dataset (De-Arteaga et al., 2019; Ravfogel et al., 2020), where biographies were scraped from the web, and annotated for the protected attribute of binary gender and target label of 28 profession classes.

Besides the binary gender attribute, we additionally consider economic status as a second protected attribute. Subramanian et al. (2021) semi-automatically labelled economic status based on the individual's home country (wealthy vs. rest of world), as geotagged from the first sentence of the biography. For bias evaluation and mitigation, we consider the intersectional groups, i.e., the Cartesian product of the two protected attributes, leading to 4 intersectional classes: female–wealthy, female–rest, male–wealthy, and male–rest.

Since the data is not directly available, in order to construct the dataset, we use the scraping scripts of Ravfogel et al. (2020), leading to a dataset with 396k biographies.[3] Following Ravfogel et al. (2020), we randomly split the dataset into train (65%), dev (10%), and test (25%).

The augmentation for economic attributes follows previous work (Subramanian et al., 2021), which results in approximate 30% instances that are labelled with both protected attributes.

| Profession | Total | Male | | Female | |
|---|---|---|---|---|---|
| | | $ | $ | $ | $ |
| professor | 21715 | 46 | 9 | 37 | 7 |
| physician | 7581 | 42 | 8 | 41 | 8 |
| attorney | 6011 | 51 | 10 | 33 | 6 |
| photographer | 4398 | 53 | 11 | 30 | 6 |
| journalist | 3676 | 41 | 9 | 41 | 9 |
| nurse | 3510 | 8 | 1 | 76 | 15 |
| psychologist | 3280 | 31 | 6 | 52 | 11 |
| teacher | 2946 | 35 | 6 | 49 | 10 |
| dentist | 2682 | 52 | 11 | 30 | 6 |
| surgeon | 2465 | 73 | 12 | 13 | 2 |
| architect | 1891 | 64 | 12 | 21 | 3 |
| painter | 1408 | 47 | 9 | 36 | 8 |
| model | 1362 | 15 | 2 | 70 | 13 |
| poet | 1295 | 46 | 7 | 39 | 8 |
| software engineer | 1289 | 70 | 14 | 14 | 2 |
| filmmaker | 1225 | 56 | 10 | 29 | 6 |
| composer | 1045 | 70 | 14 | 14 | 2 |
| accountant | 1012 | 55 | 9 | 29 | 6 |
| dietitian | 730 | 5 | 1 | 82 | 12 |
| comedian | 499 | 69 | 9 | 19 | 3 |
| chiropractor | 474 | 62 | 14 | 21 | 3 |
| pastor | 453 | 59 | 15 | 23 | 4 |
| paralegal | 330 | 12 | 3 | 70 | 15 |
| yoga teacher | 305 | 13 | 3 | 71 | 12 |
| interior designer | 267 | 16 | 4 | 67 | 12 |
| personal trainer | 264 | 41 | 10 | 42 | 7 |
| DJ | 244 | 71 | 16 | 11 | 2 |
| rapper | 221 | 75 | 15 | 9 | 1 |
| **Total** | **72578** | 9 | 45 | 7 | 39 |

Table 6: For each profession in **Bios**, the table shows the number of individuals and the breakdown across demographics as a percentage. $ and $ denote the economic status (high vs. low, respectively).

## A.3 TRUSTPILOT

We fellow previous work (Li et al., 2018) in using the **TrustPilot** dataset derived from Hovy et al. (2015), where each review is an-

---

[3]There are slight discrepancies in the dataset composition due to data attrition: the original dataset (De-Arteaga et al., 2019) had 399k instances, while 393k were collected by Ravfogel et al. (2020).

| Score | Total | Female | | | | Male | | | |
|---|---|---|---|---|---|---|---|---|---|
| | | UK | | US | | UK | | US | |
| | | Over45 | Under35 | Over45 | Under35 | Over45 | Under35 | Over45 | Under35 |
| 1 | 5051 | 13 | 17 | 3 | 3 | 23 | 31 | 4 | 6 |
| 2 | 1783 | 15 | 15 | 2 | 5 | 24 | 29 | 4 | 6 |
| 3 | 2001 | 14 | 15 | 3 | 4 | 25 | 26 | 4 | 8 |
| 4 | 4877 | 13 | 14 | 4 | 7 | 23 | 25 | 5 | 10 |
| 5 | 29489 | 17 | 11 | 5 | 4 | 26 | 22 | 5 | 9 |

Table 7: For each score in **TrustPilot**, the table shows the number of instances and the breakdown across demographics as a percentage.

notated with the target rating variable and associated with three protected labels gender (male vs. female), age (under-35-year-old vs. over-45-year-old), and, location (UK vs. the US). The original dataset contains 5 different countries (US, UK, Germany, Denmark, and France), and Li et al. (2018) discard non-English reviews after automatic language classification (Lui & Baldwin, 2012). Despite this, there are some non-English reviews in the filtered dataset, and there, we further drop instances from Germany, Denmark and France, resulting in a dataset with 54k instances in total.

## A.4 ADULT AND **COMPAS**

Except for race features, we use the same pre-processing as in Lahoti et al. (2020) for **COMPAS** (Flores et al., 2016) and **Adult** (Kohavi, 1996) datasets with 5,278 and 43,131 examples, respectively. Lahoti et al. (2020) considers binary race groups (white vs. black). However, there are more than two protected groups in the original dataset. Specifically, there are 3 race groups in **COMPAS**: African-American, Caucasian, and Other; and 5 race groups in **Adult**: White, Asian-Pac-Islander, Amer-Indian-Eskimo, Other, Black.

## B EVALUATION METRICS

Besides the absolute gap metric ($|U_{c,g} - U_c| = 0$), a broad range of formats of metrics have been introduced in previous studies to capture different assumptions about the nature of fairness. For example, Lum et al. (2022) propose to measure the variability of performance across demographic groups ($\frac{1}{G-1}\sum_g |U_{c,g} - U_c|^2$), Yang et al. (2020) only focus on the largest gap ($\max_g |U_{c,g} - U_c|$), and Feldman et al. (2015) measure performance ratio rather than gap in measuring fairness ($\frac{\max_g U_{c,g}}{\min_g U_{c,g}}$). Although, different aggregation methods have been applied to measure group-wise fairness, the optimization of any one of them is a sufficient condition for the optimization of other methods, as the optimization conditions of these metrics are identical, $U_{c,g} = U_c \forall c, g$.

For fair comparison across different debiasing approaches, we should select the hyperparameters consistently. Previous work has used different criteria for model selection, including: (1) minimum loss (Hashimoto et al., 2018; Li et al., 2018); (2) maximum utility (Lahoti et al., 2020), e.g., based on accuracy or F-measure; (3) manual selection based on visual inspection of the trade-off curve (Elazar & Goldberg, 2018; Ravfogel et al., 2020); and (4) constrained selection (Han et al., 2021b; Subramanian et al., 2021), by selecting the best fairness constrained to a particular level of performance, and vice versa. Each selection criterion reflects the performance at a particular situation, making it very hard to rigorously compare methods.

Instead, the AUC-PFC score is the integral of performance–fairness curves with respect to performance on an interval $[0, 1]$. For a particular dataset, by the definition of fairness metrics, a random classifier achieves the best fairness. Therefore, the integration from 0 to the random prediction accuracy is dataset-specific and is identical to different methods. In this paper, we normalize AUC-PFC scores for each dataset by ignoring the performance worse than random guess. Table 8 summaries the lowest accuracy scores w.r.t. each dataset.

When calculating AUC-PFC scores, for these methods that are not flexible to achieve best fairness, we manually add the random model to the calculation. Taking the Vanilla model on **Moji** as an

| Moji | Bios | TrustPilot | Adult | COMPAS |
|------|------|------------|-------|--------|
| 50% | 30% | 68% | 76% | 56% |

Table 8: Majority label proportion, i.e., lowest accuracy of each dataset.

example, the performance and fairness are $0.7109 \pm 0.0110$ and $0.6358 \pm 0.1331$, respectively. The random model corresponds to $0.5$ accuracy and $1.0$ fairness. Given these two points, the PFC is the line form $(0.7109, 0.6358)$ to $(0.5, 1)$, and the AUC-PFC score is

$$(0.7109 - 0.5) \times 0.6358 + 0.5 \times (0.7109 - 0.5) \times (1 - 0.6358) = 0.172,$$

which is consistent with Table 1.

However, we still need to select a model for early stopping before model selection. Instead of considering performance and fairness metrics separately, we use the distance to the optimal point ("DTO"), which quantifies the accuracy–fairness tradeoff (Marler & Arora, 2004; Han et al., 2022a). DTO measures the normalized Euclidean distance for a given combination of accuracy and fairness to the optimal point which denotes the ideal result, e.g., accuracy and fairness of $1.0$. It is typically unachievable in practice.

### B.1 INTERPRETATION OF AUC–PFC RESULTS

The main motivation for using AUC–PFC is for ease of comparison between approaches, as it aggregates the performance–fairness trade–off curve (PFC) of each model to a single number, enabling systematic comparison across different tasks. The two common questions related to AUC–PFC are:

- The *magnitude of AUC–PFC* differs from a single metric, and a $0.0001$ improvement in the AUC–PFC score is equivalent to a 1 percentage point (pp) improvement in both performance and fairness ($0.01 \times 0.01$). In the paper, numbers are rounded to 3 decimals, and a minimum difference in AUC–PFC ($0.001$) is roughly equivalent to a 3 pp improvement in both performance and fairness in a PFC plot.

- The *calculation of AUC–PFC* scores is normalized by the worst performance, which is the majority label proportion when using the accuracy metric. Therefore, AUC–PFC scores represent to what extent a model improves the performance or fairness over the random model.

There is no doubt that using AUC–PFC comes with certain limitations. To address the major concerns related to AUC–PFC scores, we present additional results in Appendix E, including disaggregated results for each dataset.

In particular, we provide the *PFC* of each method (e.g., Figure 6 in Appendix F), representing the best fairness that can be achieved at different performance levels, and vice versa.

One limitation of a PFC plot is that it is hard to make quantitative conclusions based on the plot itself, and we cannot conclude that one method is better than another if any intersection exists between their PFCs.

To address this problem, we additionally conduct *quantitative comparisons* across different debiasing methods by model selection w.r.t. two different criteria, and then compare both the performance and fairness of the selected models (e.g., Table 14 in Appendix E). For each method, we report the evaluation results averaged over 5 random runs with standard deviation for both the development set and test set.

As stated in Appendix F, we present disaggregated results (including a PFC plot and a table) for all 15 settings on GitHub.

## C EXPERIMENTAL DETAILS

We conduct our experiments on an HPC cluster instance with 4 CPU cores, 32GB RAM, and one NVIDIA V100 GPU.

## C.1 Assigning and smoothing proxy labels

**Assigning proxy labels**   In the current experiments, proxy labels are assigned based on the losses of each minibatch, i.e., the loss per instance is taken from a particular training iteration. We acknowledge that there are other ways of extracting training losses, e.g. taking losses from the final model or averaged over multiple iterations as the reviewer suggested, and we leave it as a future work.

**Smoothing proxy labels**   The proxy label assignment and smoothing happen simultaneously at each iteration. By doing so, our method can be incorporated into existing systems with only a few lines of changes to replace the actual protected labels with our proxy labels.

At each minibatch, the actual protected labels are replaced with smoothed proxy labels. All debiasing methods will be on the proxy labels in the later process.

During label smoothing, unsmoothed labels are used for voting to avoid inconsistency in smoothing decisions for other examples. We first collect the nearest neighbours of each instance and then do the voting for all of them.

## C.2 Models and Parameter Tuning

All approaches presented in this paper share the same dataset-specific hyperparameters as the standard model. Hyperparameters are tuned using grid-search, in order to minimize distance to the optimal.

| Hyperparameter | Search space | Best assignment | | | | |
|---|---|---|---|---|---|---|
| | | Moji | Bios | TrustPilot | Adult | COMPAS |
| number of epochs | - | 100 | | | | |
| patience | - | 10 | | | | |
| encoder | - | DeepMoji (Felbo et al., 2017) | BERT (Devlin et al., 2019) | | - | - |
| embedding size | - | 2304 | 768 | 768 | 101 | 447 |
| hidden size | - | 300 | | | | |
| number of hidden layers | *choice-integer*[1, 3] | 2 | | | | |
| batch size | *loguniform-integer*[64, 2048] | 1024 | 1024 | 1024 | 512 | 1024 |
| optimizer | - | Adam (Kingma & Ba, 2015) | | | | |
| learning rate | *loguniform-float*[$10^{-6}$, $10^{-1}$] | $3 \times 10^{-5}$ | $10^{-5}$ | $3 \times 10^{-5}$ | $3 \times 10^{-4}$ | $10^{-4}$ |
| **learning rate scheduler** | - | reduce on plateau | | | | |
| **LRS** patience | - | 2 epochs | | | | |
| **LRS** reduction factor | - | 0.5 | | | | |
| Trainable Parameter | - | 782k | 329k | 323k | 122k | 225k |

Table 9: Search space of dataset-specific hyperparameters.

All debiasing methods in this paper does not introduce extra parameter to the main task model, and will not need to considered at the inference time. As such, we provide method-specific hyperparameters separately, and the search space for method-specific hyperparameters are shared across difference datasets.

- $\text{GD}_{\text{CLA}}$ tunes the strength of the additional loss for minimizing absolute loss difference within each class. *loguniform-float*[$10^{-6}$, $10^{-1}$], 40 times.

- $\text{GD}_{\text{GLB}}$ tunes the strength of the additional loss for minimizing absolute loss difference. *loguniform-float*[$10^{-5}$, $10^{-0}$], 40 times.

- FairBatch tunes the adjustment rate for resampling probabilities. *loguniform-float*[$10^{-4}$, $10^{-0}$], 40 times.

- Adv tunes the weights of unlearning adversaries in training. *loguniform-float*[$10^{-2}$, $10^{+2}$], 40 times.

- SemiAdv tunes the weights of unlearning adversaries in training. *loguniform-float*[$10^{-2}$, $10^{+2}$], 40 times.

| Dataset | Moji | Bios | | | TrustPilot | | | | Adult | | | COMPAS | | |
|---|---|---|---|---|---|---|---|---|---|---|---|---|---|---|
| Attribute | $R$ | $G$ | $E$ | $G{\times}E$ | $G$ | $A$ | $C$ | $G{\times}A{\times}C$ | $G$ | $R$ | $G{\times}R$ | $G$ | $R$ | $G{\times}R$ |
| Vanilla | 58 | 47 | 37 | 39 | 20 | 20 | 20 | 18 | 25 | 17 | 22 | 9 | 10 | 10 |
| $GD_{CLA}$ | 55 | 131 | 131 | 149 | 43 | 36 | 36 | 42 | 26 | 26 | 27 | 10 | 11 | 12 |
| $GD_{GLB}$ | 54 | 121 | 116 | 157 | 34 | 33 | 33 | 45 | 27 | 28 | 28 | 10 | 12 | 11 |
| FairBatch | 54 | 46 | 48 | 47 | 26 | 26 | 27 | 24 | 24 | 21 | 21 | 11 | 10 | 11 |
| Adv | 47 | 54 | 48 | 41 | 29 | 28 | 28 | 23 | 23 | 29 | 29 | 11 | 9 | 9 |
| SemiAdv | 51 | 57 | 53 | 42 | 30 | 29 | 30 | 25 | 24 | 20 | 29 | 11 | 11 | 9 |
| ARL | 47 | 71 | 73 | 40 | 27 | 25 | 27 | 20 | 22 | 21 | 23 | 11 | 11 | 12 |
| $ULPL{+}GD_{CLA}$ | 148 | 35 | 37 | 40 | 31 | 33 | 34 | 33 | 51 | 55 | 54 | 15 | 16 | 18 |
| $ULPL{+}GD_{GLB}$ | 148 | 35 | 37 | 38 | 32 | 33 | 34 | 32 | 49 | 54 | 54 | 15 | 16 | 17 |
| ULPL+Adv | 179 | 49 | 50 | 41 | 47 | 46 | 46 | 36 | 62 | 61 | 68 | 15 | 16 | 17 |

Table 10: Average computational budget, measured in seconds.

- ARL tunes the learning rate of learning adversaries in training. *loguniform-float*$[10^{-4}, 10^{+2}]$, 40 times.
- ULPL methods tunes the $k$ from 1 to 15, and $p$-norm from 2 to 6.

Notice that, this paper report the AUC-PFC, which eliminate the requirement for model selection, i.e., there is no best-found trade-off hyperparameters w.r.t. bias mitigation.

### C.3 COMPUTATIONAL BUDGET

Table 10 shows average GPU time of model training. Noticing that debiasing components will not be used for inference, i.e., different methods have identical inference cost.

### C.4 PARAMETER TUNING FOR LABEL SMOOTHING

For **Bios**, class-specific neighbourhood smoothing degrades to naive proxy labels when there is only a small number of instances in a particular class. For example, there are 28 distinct target classes in the **Bios** dataset, with a highly skewed distribution. As such, there can be only one instance per target class in a minibatch, and the neighbourhood smoothing does not work in this case.

For **TrustPilot**, we hypothesise that it is due to the leakage of protected information being very low, and accordingly the neighbourhoods of instances being noisy.

**The selection of $k$ for label smoothing.** As observed in Section 5.2, the optimal value of $k$ varies greatly across datasets and debiasing methods, and in the context of this paper, we deal with this through a simple grid search over different values of $k$, which is computationally expensive.

Although we do not currently have an algorithm for efficiently optimizing $k$ at this time, we have observed that the value of $k$ is positively correlated with model leakage and unfairness. Therefore, we could start tuning the value of $k$ from a large value if the model is significantly biased, as the instances from the same protected group are likely to be close to each other. Otherwise, we can use the proxy labels without smoothing if the results are reasonably fair.

## D THEORETICAL JUSTIFICATION

### D.1 FROM EMPIRICAL LOSSES TO UTILITY METRICS

For illustration purposes, we assume binary settings for both target class and protected attribute labels. In Section 3.3, we have shown that the proposed method can be used to improve the equal opportunity fairness.

**Demographic parity (DP)** For DP fairness, the predictions are expected to be independent from protected attributes ($\hat{y} \perp z$), and the fairness is satisfied if the differences in positive prediction rate

between demographic groups are zero: $p(\hat{y} = 1|z = 0) = p(\hat{y} = 1|z = 1)$. Thus,

$$p(\hat{y} = 1, y = 0|z = 0) + p(\hat{y} = 1, y = 1|z = 0)$$
$$= p(\hat{y} = 1, y = 0|z = 1) + p(\hat{y} = 1, y = 1|z = 1).$$

Since $p(\hat{y} = 0, y|z) + p(\hat{y} = 1, y|z) = 1, \forall y, z$, by replacing $p(\hat{y} = 1, y = 0|z)$ with $1 - p(\hat{y} = 0, y = 0|z)$,

$$(1 - p(\hat{y} = 0, y = 0|z = 0)) + p(\hat{y} = 1, y = 1|z = 0)$$
$$= (1 - p(\hat{y} = 0, y = 0|z = 1)) + p(\hat{y} = 1, y = 1|z = 1).$$

An equivalent condition to the DP fairness is that

$$p(\hat{y} = 1, y = 1|z = 0) - p(\hat{y} = 0, y = 0|z = 0)$$
$$= p(\hat{y} = 1, y = 1|z = 1) - p(\hat{y} = 0, y = 0|z = 1).$$

A sufficient condition for DP is, both $p(\hat{y} = 1, y = 1|z = 0) = p(\hat{y} = 1, y = 1|z = 1)$ and $p(\hat{y} = 0, y = 0|z = 0) = p(\hat{y} = 0, y = 0|z = 1)$ are satisfied. Next, we show how to map the conditional joint probability to training losses. As for the $y = 1$, recall that, $\mathcal{L}^1$ is an unbiased estimator of $-\log(p(\hat{y} = 1|y_i = 1)$ (Equation (1)),

$$\mathcal{L}^1 = -\log(p(\hat{y} = 1|y_i = 1)$$
$$\mathcal{L}^1 = -\log(p(\hat{y} = 1|y_i = 1) - \log(p(y_i = 1)) + \log(p(y_i = 1))$$
$$\mathcal{L}^1 = -\log(p(\hat{y} = 1|y_i = 1)p(y_i = 1)) + \log(p(y_i = 1))$$
$$\mathcal{L}^1 = -\log(p(\hat{y} = 1, y_i = 1)) + \log(p(y_i = 1))$$

By substituting the joint probability with losses,

$$p(\hat{y} = 1, y = 1|z = 0) = p(\hat{y} = 1, y = 1|z = 1)$$
$$-\log(p(\hat{y} = 1, y = 1|z = 0)) = -\log(p(\hat{y} = 1, y = 1|z = 1))$$
$$\mathcal{L}^{1,0} - \log(p(y_i = 1|z_i = 0)) = \mathcal{L}^{1,1} - \log(p(y_i = 1|z_i = 1))$$
$$\mathcal{L}^{1,0} - \mathcal{L}^{1,1} = \log(\frac{p(y_i = 1|z_i = 0)}{p(y_i = 1|z_i = 1)})$$

Similarly, $\mathcal{L}^0$ is an unbiased estimator of $-\log(p(\hat{y} = 0, y_i = 0) + \log(p(y_i = 0))$, and the DP condition for $y = 0$, $p(\hat{y} = 1, y = 1|z = 0) = p(\hat{y} = 1, y = 1|z = 1)$, is equivalent to $\mathcal{L}^{0,0} - \mathcal{L}^{0,1} = \log(\frac{p(y_i = 0|z_i = 0)}{p(y_i = 0|z_i = 1)})$.

Notice that $p(y|z), \forall y, z$ are dataset-specific constant numbers, and if $p(y|z = 0) = p(y|z = 1), \forall y$, the DP conditions are identical to Equalized Odds fairness (Hardt et al., 2016), and can be approximated by $\mathcal{L}^{1,0} = \mathcal{L}^{1,1}$ and $\mathcal{L}^{0,0} = \mathcal{L}^{0,1}$. Last but not least, recall that the nature of DP assumes $y$ and $z$ are independent, therefore, $p(y|z = 0) - p(y|z = 1) \approx 0, \forall y$ generally holds when the DP fairness is desired.

**Confusion-matrix based metrics**   So far, we have shown that minimizing loss differences can approximate the optimization of the two most wildly used notions of fairness: EO and DP fairness. Since model predictions and target labels are observed during training, such approximation can also be applied to other confusion-matrix-based metrics. For example, the cross-entropy loss of instances w.r.t. predictions as 0 and 1 are approximations of the positive predictive value ($p(y = 1|\hat{y} = 1)$) and negative predictive value ($p(y = 0|\hat{y} = 0)$), respectively.

## D.2 Optimizing fairness with proxy labels

**Bias mitigation for different fairness criteria** Overall, ULPL only assigns proxy labels to training instances, and the optimization for fairness is achieved using existing supervised debiasing methods, by learning uniform representations across proxy groups (such as Adv) or minimizing loss disparities in training (such as $GD_{CLA}$ and $GD_{GLB}$). Based on ULPL, different fairness criteria can be optimized by employing different variants of a particular debiasing method. Taking Adv as an example, a discriminator is trained to recover the protected information from hidden representations, and the main task model is optimized to remove protected information from hidden representations through unlearning the discriminator. By doing so, the hidden representations and corresponding predictions are expected to be independent of the protected attribute, ensuring DP fairness. To adopt Adv for EO fairness, the discriminator takes target labels into consideration, e.g. training a specific discriminator for instances with positive target class only, and the removal of protected information is then class-dependent, aligning with the definition of EO fairness.

# E  Additional Results

## E.1  Proxy Label Assignment – Moji

| Sentiment | Race | $P(z'=1)$ | PPR↑ | TPR↑ | FPR↓ | PPV↑ | NPV↓ |
|---|---|---|---|---|---|---|---|
| **sad** | SAE | $17.0\pm1.2$ | $79.9\pm0.8$ | $91.2\pm0.7$ | $35.0\pm3.1$ | $91.3\pm0.7$ | $64.8\pm2.2$ |
|  | AAE | $68.5\pm1.7$ | $14.3\pm0.7$ | $45.7\pm2.6$ | $6.4\pm0.8$ | $64.2\pm3.2$ | $87.3\pm0.6$ |
| **happy** | SAE | $55.3\pm3.0$ | $20.1\pm0.8$ | $65.0\pm3.1$ | $8.8\pm0.7$ | $64.8\pm2.2$ | $91.3\pm0.7$ |
|  | AAE | $15.8\pm2.3$ | $85.7\pm0.7$ | $93.6\pm0.8$ | $54.3\pm2.6$ | $87.3\pm0.6$ | $64.2\pm3.2$ |

Table 11: Proxy label assignment without smoothing and evaluations for the Vanilla model over **Moji**. Evaluation results $\pm$ standard deviation (%) are averaged over 5 runs with different random seeds. $\pm P(z'=1)$ refers to the proportion of instances being assigned with 1, indicating worse-performed groups. Evaluating metrics include: (1) positive predictive rate (PPR), corresponding to the demographic parity fairness (Blodgett et al., 2016), (2) true positive rate (TPR) and false positive rate (FPR), corresponding to equalized odds and equal opportunity fairness (Hardt et al., 2016), and (3) positive predictive value (PPV) and negative predictive value (NPV), corresponding to test fairness (Chouldechova, 2017).

We first investigate if the ULPL labels are meaningful through the lens of training examples in the **Moji** dataset. Table 11 presents the results of the Vanilla model.

It can be seen that, AAE tweets are more likely to be classified as happy, while SAE tweets are more likely to be classified as sad, resulting in consistent trend in gaps with respect to PPR, TPR, FPR, PPV, and NPV. Based on loss-disparities, AAE instances with sad target labels are more possible to be assigned with $z'=1$ (68.5% vs. 17% for SAE and AAE, respectively), encouraging debiasing methods to focus more on sad-AAE instances in training. Similarly, happy-SAE instances are more likely to be assigned with $z'=1$, indicating that happy-SAE are upweighted in training.

For the dataset distribution perspective of view, as introduced in Appendix A.1, **Moji** is balanced with respect to both sentiment and ethnicity but skewed in terms of sentiment–ethnicity combinations (40% happy-AAE, 10% happy-SAE, 10% sad-AAE, and 40% sad-SAE, respectively), which is closely related to the DL assignments that minority groups within each target class are assigned with $z'=1$. I.e., our proposed proxy label differentiates minority groups with majority groups within each target class without observing demographic labels.

## E.2  Protected label predictability after debasing

Neighbour smoothing requires protected information in hidden representations during the whole training, requiring encoded protected information in hidden representations. Han et al. (2021b) show that although supervised debiasing methods have shown success in reducing performance disparities in downstream tasks, the predictability of protected attributes in debiased hidden representations is

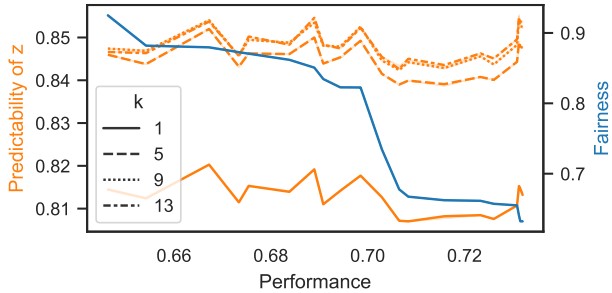

Figure 4: Predictability after debiasing.

still well above the ideal value. We take ULPL+GD$_{\text{CLA}}$ as an example and explore the protected label predictability across different debiased models (i.e., different trade-offs).

As seen from Figure 4, fairness scores (the blue line) improve at the cost of performance. However, the predictability of protected labels is quite stable at a high level, indicating that protected information is still encoded in debiased models, and our proposed neighbourhood smoothing method is robust to bias mitigation.

### E.3 EFFECTIVENESS OF THE NEIGHBOUR SMOOTHING

In smoothing z′ labels, we hypothesis that the nearest neighbours of an instance are likely to from the same protected group. Except the instance itself, the remaining nearest neighbours are essentially the results of a standard $k$-nearest-neighbour ($k$-NN) model. Therefore, we perform analysis based on standard $k$-NN models, and investigate if the remaining nearest neighbours are helpful for label smoothing, i.e., from the same protected group as the target instance.

### E.4 OTHER FAIRNESS METRICS, DP

We investigate the robustness of ULPL methods to other notions of fairness. For illustration purposes, we focus on demographic parity fairness (DP) (Blodgett et al., 2016) in this experiment, which requires model predictions to be independent with protected attributes. Again, we aggregate accuracy and DP fairness trade-offs as the AUC-PFC scores. Since DP is sensitive to class imbalance and there is no standard way of generalizing DP to multi-class classification tasks, we only conduct experiments over binary classification tasks, including **Moji**, **Adult**, and **COMPAS**. Table 12 shows results w.r.t. demographic parity fairness.

Trends for different methods are similar to the results of equal opportunity fairness, indicating that debasing methods are robust to different fairness metrics.

| Dataset | Moji | Adult | | | COMPAS | | |
|---|---|---|---|---|---|---|---|
| Attribute | R | G | R | G×R | G | R | G×R |
| Vanilla | 0.173 | 0.087 | 0.076 | 0.073 | 0.109 | 0.097 | 0.094 |
| GD$_{\text{CLA}}$ | 0.253 | 0.091 | 0.081 | 0.079 | 0.121 | 0.094 | 0.085 |
| GD$_{\text{GLB}}$ | 0.233 | 0.091 | 0.081 | 0.077 | 0.121 | 0.098 | 0.094 |
| FairBatch | 0.254 | 0.091 | 0.079 | 0.073 | 0.117 | 0.103 | 0.094 |
| Adv | 0.254 | 0.091 | 0.077 | 0.074 | 0.117 | 0.094 | 0.945 |
| SemiAdv | 0.255 | 0.090 | 0.077 | 0.074 | 0.119 | 0.093 | 0.945 |
| ARL | 0.197 | 0.092 | 0.081 | 0.078 | 0.120 | 0.092 | 0.092 |
| ULPL+GD$_{\text{CLA}}$ | 0.214 | 0.090 | 0.082 | 0.080 | 0.123 | 0.101 | 0.095 |
| ULPL+GD$_{\text{GLB}}$ | 0.192 | 0.090 | 0.081 | 0.078 | 0.124 | 0.100 | 0.097 |
| ULPL+Adv | 0.186 | 0.091 | 0.083 | 0.080 | 0.122 | 0.099 | 0.092 |

Table 12: AUC-PFC based on demographic parity fairness.

| Dataset | S | T | Vanilla | GD$_{\text{CLA}}$ | FairBatch | SemiAdv | ARL | ULPL+GD$_{\text{CLA}}$ |
|---|---|---|---|---|---|---|---|---|
| **Bios** | G | E | −0.0011 | 0.0003 | −0.0054 | −0.0021 | 0.0018 | 0.0002 |
| | E | G | −0.0041 | −0.0131 | −0.0166 | −0.0188 | −0.0028 | −0.0120 |
| **TrustPilot** | G | A | 0.0025 | 0.0021 | −0.0007 | −0.0018 | −0.0018 | 0.0006 |
| | G | C | 0.0047 | 0.0012 | −0.0010 | −0.0015 | 0.0004 | 0.0008 |
| | A | G | −0.0017 | −0.0032 | −0.0006 | 0.0001 | 0.0009 | −0.0020 |
| | A | C | 0.0018 | −0.0009 | −0.0001 | −0.0000 | 0.0012 | −0.0006 |
| | C | G | −0.0041 | −0.0023 | −0.0001 | −0.0003 | −0.0007 | −0.0023 |
| | C | A | −0.0027 | 0.0011 | 0.0011 | −0.0008 | −0.0023 | −0.0010 |
| **Adult** | G | R | 0.0053 | −0.0098 | −0.0045 | 0.0037 | −0.0080 | −0.0060 |
| | R | G | −0.0023 | −0.0010 | −0.0040 | −0.0033 | −0.0012 | −0.0020 |
| **COMPAS** | G | R | −0.0012 | −0.0036 | −0.0036 | 0.0002 | −0.0014 | 0.0011 |
| | R | G | 0.0058 | −0.0026 | 0.0025 | −0.0064 | −0.0083 | −0.0035 |
| **Average** | | | 0.0002 | −0.0026 | −0.0027 | −0.0026 | −0.0019 | −0.0022 |

Table 13: AUC-PFC score differences between debiasing w.r.t. target protected attributes (**T**), and source protected attributes (**S**). Larger numbers indicate better generalizations to unobserved protected groups.

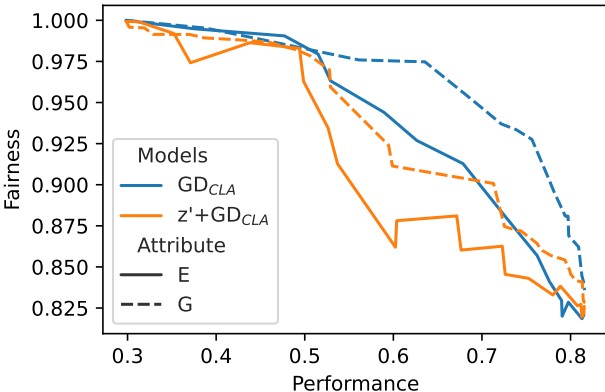

Figure 5: PFC of in-domain and cross-domain debiasing over the **Bios**–gender dataset.

### E.5 How does bias mitigation affect fairness for unobserved groups?

Proxy group labels ($z'$) are dynamically adjusted at each minibatch during training, which differs from fixed protected labels in supervised debiasing. As a result, supervised debiasing methods based on the observed protected attributes $z$ mitigate biases for particular protected groups.

While the proposed proxy label approaches focus on the group of instances that are underrepresented during training, which is expected to be more general than debiasing to a particular protected attribute. Figure 5 demonstrates the difference in AUC-PFC scores between in-domain debiasing and cross-domain debiasing. For each debiasing method, we train the debiased model and conduct model selections based on the source protected attribute (Economy). The trained models are then evaluated w.r.t. the target unobserved protected attribute (Gender). The ability to generalize to unobserved protected attributes is measured as the difference between in-domain and cross-domain AUC-PFC.

Table 13 summaries AUC-PFC differences across difference datasets w.r.t. a subset of debiasing methods. Overall, Vanilla shows the best generalization to unobserved protected attributes, and unsupervised debiasing methods are better than supervised and semi-supervised debiasing methods.

Intuitively, the Vanilla model and unsupervised debiasing methods (ARL and ULPL+GD$_{\text{CLA}}$) are agnostic to cross-attribute generalization, and their difference should be close to 0 which is clearly not the case. The AUC-PFC decrease of these methods is caused by the model selection, i.e., although the training process is identical, different models may be selection based on their own pro-

tected attributes. As shown in Figure 5, the selected model of ULPL+GD$_{\text{CLA}}$ based on economy labels at accuracy around 0.6 is not a Pareto point for gender, confirming that decreases in AUC-PFC of Vanilla and unsupervised debiasing methods are caused by cross-domain model selection.

### E.6 THE APPLICATION OF ULPL TO DIFFERENT DEBIASING APPROACHES

GD$_{\text{CLA}}$ are representatives of debiasing methods that directly optimize loss parity. In particular, the training objective of GD$_{\text{CLA}}$ is:

$$\mathcal{L}_{\text{GD}_{\text{CLA}}} = \mathcal{L} + \lambda_{\text{GD}_{\text{CLA}}} \sum_{\text{c}} \sum_{\text{g}} |\mathcal{L}^{\text{c,g}} - \mathcal{L}^{\text{c}}|,$$

where $|\mathcal{L}^{\text{c,g}} - \mathcal{L}^{\text{c}}|$ is optimized to achieve better fairness. Since $\mathcal{L}^{\text{c,g}}$, $\mathcal{L}^{\text{c}}$, and $\mathcal{L}$ are average losses, their magnitude are irrelevant to subset sizes ($n_{\text{c,g}}$, $n_{\text{c}}$, and $n$, respectively), which in turn applies the same strength of fairness regularization to all subset of instances $\mathcal{D}_{\text{c,g}}, \forall \text{c, g}$. In other words, GD$_{\text{CLA}}$ ignores the influence of group size in bias mitigation, resulting in robustness to imbalanced class distributions.

There is a perfect alignment between ULPL and GD$_{\text{CLA}}$ in the sense that the proxy group $z' = 1$ will always be upweighted during the optimization of GD$_{\text{CLA}}$. I.e., ULPL+GD$_{\text{CLA}}$ reduces the loss disparities across instances within each target class, which in turn improves the lower bound of group-wise fairness, especially for the EO fairness. As a result of the consistency, we observe that ULPL+GD$_{\text{CLA}}$ outperforms other unsupervised methods.

GD$_{\text{GLB}}$ is a variant of GD$_{\text{CLA}}$, and they can only be differentiated by the way of incorporating fairness regularization:

$$\mathcal{L}_{\text{GD}_{\text{GLB}}} = \mathcal{L} + \lambda_{\text{GD}_{\text{GLB}}} \sum_{\text{c}} \sum_{\text{g}} |\mathcal{L}^{\text{c,g}} - \mathcal{L}|,$$

where the average loss in the regularization term is based on all instances ($\mathcal{L}$), differing from the average loss within each target class ($\mathcal{L}^{\text{c}}$) for GD$_{\text{CLA}}$. As a result, GD$_{\text{GLB}}$ additionally encourages the performance parity across target class, which is typically known as long-tail learning. However, ULPL+GD$_{\text{GLB}}$ could potentially lead to worse results for better-performed target classes. For example, for $i$th target class, assuming that loss differences have been minimized (i.e., $\mathcal{L}^{i,z'=0} \approx \mathcal{L}^{i,z'=1}$), it is possible that all instances with target label $y = i$ will be under-fitted if $\mathcal{L}^{i} > \mathcal{L}$.

Adv represents a different family of debiasing methods which aims at learning fair hidden representations. The training objective of Adv includes the mutual information ($MI$) to the training objective in addition to standard loss:

$$\mathcal{L}_{\text{Adv}} = \mathcal{L} + MI(z, \boldsymbol{h}),$$

where the $\boldsymbol{h} = e(\boldsymbol{x})$ is the hidden representation of input $\boldsymbol{x}$ extracted from the encoder $e$. By minimizing the $MI$ objective, we expected the learned hidden representations $\boldsymbol{h}$ are orthogonal to protected attributes z.

In practice, $MI(z, \boldsymbol{h})$ is expressed by the combination of marginal entropy ($H(z)$) and conditional entropy ($H(z|\boldsymbol{h})$): $MI(z, \boldsymbol{h}) = H(z) - H(z|\boldsymbol{h})$, where $H(z)$ is a constant number and can be ignored in the optimization, and $H(z|\boldsymbol{h})$ is estimated by an adversary ($d$) that is trained to identify the protected attributes ($\hat{z} = d(\boldsymbol{h})$).

The key step of Adv is the training of $d$, i.e., if $d$ can effectively recover $z$ from $h$. One problem associated with ULPL is that the mapping from proxy labels to ground truth labels is class-specific, making it harder for the recovering. Therefore, although the adversaries are non-linear classifiers, the effectiveness of ULPL+Adv is not as good as other ULPL $+*$ methods as shown in Table 1.

## F FULL RESULTS

In addition to AUC-PFC scores, we present PFC and evaluation results for each dataset. We investigate two different selection criteria and report the evaluation results over the development and test sets. Specifically, we conduct model selection over the development set based on: (1) maximum fairness within a performance trade-off threshold of 5% (**F@P−5%**); and (2) maximum fairness within a performance trade-off threshold of 10% (**F@P−10%**).

| Selection | Method | Test Set | | Development Set | |
|---|---|---|---|---|---|
| | | Performance | Fairness | Performance | Fairness |
| **F@P-5%** | Vanilla | 71.1± 1.1 | 63.6± 1.3 | 71.4± 1.2 | 65.9± 1.2 |
| | $GD_{CLA}$ | 74.2± 0.3 | 92.9± 1.4 | 73.7± 0.3 | 94.5± 1.0 |
| | $GD_{GLB}$ | 75.9± 0.4 | 77.6± 1.3 | 75.2± 0.2 | 79.7± 1.6 |
| | FairBatch | 75.4± 0.4 | 90.3± 0.8 | 74.8± 0.2 | 90.9± 0.7 |
| | Adv | 75.3± 0.4 | 89.7± 1.6 | 74.9± 0.3 | 91.0± 0.8 |
| | SemiAdv | 75.7± 0.2 | 90.1± 0.5 | 74.9± 0.4 | 91.1± 0.5 |
| | ARL | 70.7± 1.0 | 78.5± 5.5 | 69.2± 1.0 | 78.1± 5.7 |
| | ULPL+RL | 66.3± 2.0 | 85.7± 4.9 | 66.1± 2.1 | 86.7± 4.6 |
| | ULPL+$GD_{CLA}$ | 66.7± 2.6 | 88.0± 7.0 | 66.2± 2.4 | 88.8± 6.8 |
| | ULPL+$GD_{GLB}$ | 72.6± 0.4 | 65.1± 1.6 | 72.5± 0.6 | 66.9± 1.3 |
| | ULPL+Adv | 67.8± 5.6 | 69.4± 9.2 | 68.2± 5.5 | 71.2± 8.9 |
| **F@P-10%** | Vanilla | 71.1± 1.1 | 63.6± 1.3 | 71.4± 1.2 | 65.9± 1.2 |
| | $GD_{CLA}$ | 74.2± 0.3 | 92.9± 1.4 | 73.7± 0.3 | 94.5± 1.0 |
| | $GD_{GLB}$ | 75.9± 0.4 | 77.6± 1.3 | 75.2± 0.2 | 79.7± 1.6 |
| | FairBatch | 75.4± 0.4 | 90.3± 0.8 | 74.8± 0.2 | 90.9± 0.7 |
| | Adv | 75.3± 0.4 | 89.7± 1.6 | 74.9± 0.3 | 91.0± 0.8 |
| | SemiAdv | 75.7± 0.2 | 90.1± 0.5 | 74.9± 0.4 | 91.1± 0.5 |
| | ARL | 63.7± 3.8 | 84.6± 4.2 | 63.0± 3.9 | 85.9± 3.5 |
| | ULPL+$GD_{CLA}$ | 64.6± 3.3 | 92.5± 3.0 | 64.8± 2.2 | 92.6± 2.4 |
| | ULPL+$GD_{GLB}$ | 72.6± 0.4 | 65.1± 1.6 | 72.5± 0.6 | 66.9± 1.3 |
| | ULPL+Adv | 67.8± 5.6 | 69.4± 9.2 | 68.2± 5.5 | 71.2± 8.9 |

Table 14: Evaluation results ± standard deviation (%) of selected models over the **Moji** dataset.

For the demonstration purpose, here we present the results for **Moji**. The full disaggregated results of 15 settings can also be seen at `https://github.com/HanXudong/An_Unsupervised_Locality-based_Method_for_Bias_Mitigation/blob/main/unsupervised_bias_mitigation/NB_Appendix_indomain_tradeoffs_dispaly.ipynb`.

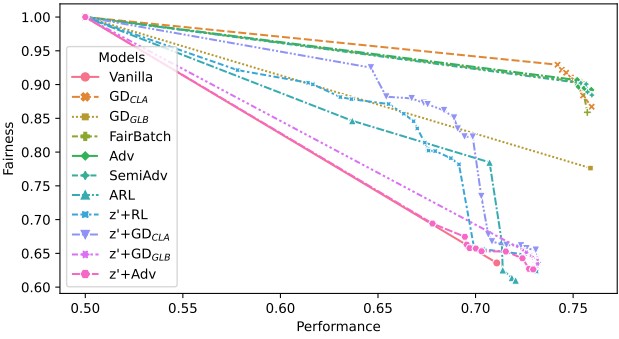

Figure 6: PFC over the **Moji** dataset.

**Moji**  Figure 6 shows performance–fairness trade-off curves (PFC) of each debiasing method. Table 14 summarises performance and equal opportunity fairness results w.r.t. two different selection criteria over the test set and dev set.

The areas under each PFC in Figure 6 correspond to a number in Table 1. Consistent with the Table 1, it can been from Figure 6 that $GD_{CLA}$ results in the best PFC. In addition, PFCs of Adv and SemiAdv are highly overlapped with each other, and their AUC–PFC scores are also identical in Table 1. Last but not least, ULPL+$GD_{CLA}$ is better than ARL most of the time in Figure 6, which is summarized as a 0.016 improvement in terms of AUC-PFC score in Table 1.

