# OpenReview forum: "Everybody Needs Good Neighbours: An Unsupervised Locality-based Method for Bias Mitigation"
_ICLR.cc/2023/Conference — ICLR 2023 poster_

### Official Review · Reviewer_y5Cm · 2022-10-21

**Confidence:** 4
**Correctness:** 3
**Technical Novelty And Significance:** 3
**Empirical Novelty And Significance:** 3
**Recommendation:** 6

**Clarity, Quality, Novelty And Reproducibility:**

Clarity: Overall, the work is well presented. The method and experimental setup are well described.
Novelty: Some works follow a similar high-level sprit (e.g, Hashimoto et al.) but the follow-up ARL method is a competitor in the experiments and an improvement over this method is shown. Therefore, I would consider this work as sufficiently novel, but there might be some works which I have missed.
Reproducibility. The authors provide the full source code which appears to be in a good shape. Therefore, I am confident that the results are reproducible.


**Strength And Weaknesses:**

Strengths:

1. The paper is well written. I enjoyed reading this paper and found it well-structured and good to follow.
2. The paper addresses an important and timely problem. Bias is common in machine learning and methods to deal with it are relevant in practice. Furthermore, bias is often introduced implicitly, i.e., without group labels known at model training time. Therefore, the proposed method may be a valuable step towards fairer automated decisions.
3. The experimental evaluation is extensive and rigorous and overall convincing. It includes several data modalities (tabular and text), fairness measures, datasets and baselines. Furthermore, important sanity checks such as the validity of the nearest-neighbor approximation for smoothing are in place. I briefly looked at the experimental code which appears to be in a good state. Intersectional discrimination of multiple attributes is also considered.
4. The approach is simple and can be implemented in a few lines of code. Also, it only has one hyperparameter (the number of neighbors used for smoothing). This makes the method ready-to-use for practical applications.

Weaknesses:

1. There is no comparison between proxy labels and ground truth groups (e.g., given by race, gender, etc.). However, it is frequently claimed that the proxy labels correlate with demographic groups (for instance, right in the abstract, it says: “[...] that correlate with unknown demographic data”). The claim is repeated in 5.1, however, the performance gap is compared to the proxy labels only and not the ground truth group labels. This could be achieved by providing the statistics, e.g., confusion matrices, with ground truth group labels.
2. The theoretical grounding of the work is weak, which raises the question whether there exist other simpler and principled solutions. While the paper provides some intuition for why the proposed method works, it lacks theoretical evidence from point to point. For instance, it is not clear why U should be generally replaced with the Loss in Section 3.2 and how the group-based formulation is transformed to an individual one. I see that the experimental results are promising, but the derivation could be backed with more evidence or theoretical results.
3. There are no theoretical guarantees that the proposed method will lead to fairer models and that the proxy objective corresponds to the actual one (or bounds on the error). This also raises the question if the fairness could possibly be decreased. This could be a problem in high-stakes scenarios. In such settings, a black-box method like the one presented is unlikely to be deployed.

Minor comments:
1. I would have appreciated further evidence or intuition for some claims made in the paper. Here are some examples:
(1) Because TrustPilot is already fair, the ground truth labels cannot be aligned with the TPR gaps (Section 5.1)
(2) Why is GD-CLA more robust to conflated classes (final paragraph of 4.2)
(3) The discussion of upstream and downstream debiasing (Section 2, “Unsupervised bias mitigation”). As far as I understand, upstream debiasing is also unsupervised? Thus, a comparison with one of these methods could be insightful in terms of future work, since these methods could be applicable to the same set of problems.
2. Related work: Please use \citep (end of first paragraph, Point (1) in second paragraph).

Questions:
1. How exactly was the ROC-RFC computed? Were the scores normalized? For instance, looking at Figure 5b), the area under the curve is well above 0.8 x 0.8=0.64 for both methods. Even if you normalize the curve (taking the left and right-most point as references), the areas are well above 0.5. However, in Table 1, values reported for Bios-G are all below 0.5. How can this be explained?
2. ROC-PFC values for Trust Pilot, Adult, Compas are extremely low (~0.1). This indicates that performance is hurt drastically even with slowly introduced fairness. Why is this the case? Is the AUC-PFC a still a representative measure in this case? Maybe this is connected to the first question.
3. Table 3 shows the best values for the nearest neighbors smoothing which are different across datasets. How would you choose a suitable value for k in practice, without having access to ground truth fairness measures?





**Summary Of The Paper:**

The paper proposes an unsupervised method for bias-mitigation. The presented approach leverages proxy labels derived from comparing the instance-wise loss values to a threshold. These proxy labels are consolidated/smoothed through nearest neighbors in latent space (through embedding). The labels indicate poorly and well represented groups, which can be used in combination with arbitrary supervised methods for obtaining fairer classification models. The approach is extensively evaluated and is competitive with group-label supervised methods for bias mitigation.

**Summary Of The Review:**

The topic of bias mitigation is relevant and the empirical results in this paper are interesting. While the empirical evidence suggests that the method improves fairness on real datasets, there are no theoretical guarantees or considerations, making it hard to deploy this method in high-stakes applications. However, I found the experiments well-designed and executed and the results very promising, which is why I am currently leaning towards acceptance.

---

> ### Author Response · Authors · 2022-11-18
> **Response to Reviewer y5Cm (1/2)**
>
> We thank Reviewer y5Cm for the detailed comments and thoughtful feedback on our work. Responses to specific comments are provided below.
>
> ## Q4.1 Comparison between proxy labels and ground truth groups.
> In Section 5.1, the signed performance gap is computed based on the ground truth group labels (${\textnormal{z}}$). Therefore, the strong correlation between the signed performance gap and the proxy label (${\textnormal{z}}'$) implies a strong correlation between ${\textnormal{z}}$ and ${\textnormal{z}}'$.
> Put another way, the performance gap can be treated as a mapping of group truth group labels within each target class, such that the meaning of ground truth labels and protected labels is comparable.
>
> For example, assume that the target label (${\textnormal{y}}$) and ${\textnormal{z}}$ are binary, and consider worst-case fairness, e.g. $p(\hat{{\textnormal{y}}} = 1, {\textnormal{y}}=1 | {\textnormal{z}} = 0) \approx 0$ and $p(\hat{{\textnormal{y}}} = 1, {\textnormal{y}}=1 | {\textnormal{z}} = 1) \approx 1$, where the TPR gap between the two groups is 1. Within the negative class (${\textnormal{y}}=0$), instances with ${\textnormal{z}}=0$ will be assigned with proxy label 1 ${\textnormal{z}}'=1$. Similarly, within the positive class (${\textnormal{y}}=1$), instances with ${\textnormal{z}}=0$ will be assigned with proxy label 0 ${\textnormal{z}}'=0$.
> I.e., $p({\textnormal{z}}' = 1 | {\textnormal{z}} = 0, {\textnormal{y}} = 0) = p({\textnormal{z}}' = 0 | {\textnormal{z}} = 0, {\textnormal{y}} = 1) = 1$. Clearly, there is a strong correlation between the ground truth label ${\textnormal{z}}$ and the proxy label ${\textnormal{z}}'$. However, assuming $p({\textnormal{y}} = 0) = p({\textnormal{y}} = 1)$, there is a weak correlation between ${\textnormal{z}}$ and ${\textnormal{z}}'$ if we ignore the class information, $p({\textnormal{z}}' = 1 | {\textnormal{z}} = 0) = p({\textnormal{z}}' = 0 | {\textnormal{z}} = 0) = 0.5$. To address this problem and avoid manually mapping ${\textnormal{z}}$, we measure the correlation between ${\textnormal{z}}'$ and ${\textnormal{z}}$-based signed gaps, reflecting the correlation between ${\textnormal{z}}'$ and ${\textnormal{z}}$ within all target classes.
>
>
> ## Q4.2 Theoretical justification.
> Please refer to the theoretical justification (**Q0.1**) in the general response. We briefly summarize responses to specific questions below.
>
> We show that losses within each class can be unbiased estimators of TPR and TNR, and confusion-matrix-based metrics can be reformulated as their combinations. For example, precision (PPV) = TP/(TP+FP), where TP = TPR$\times$P, FP=(1-TNR)$\times$P, and P and N are the number of instances with positive and negative target classes, respectively.
>
> Moreover, we show that minimizing loss disparities is sufficient for minimizing performance disparities, and using proxy labels for bias mitigation improves the lower bound of fairness.
>
> ## Q4.3 Further evidence and intuition.
>
> For the TrustPilot dataset, we agree that the models are already quite fair compared with other datasets, and as a result, their proxy labels cannot be correctly assigned, and label smoothing is counter-productive.
>
> For the discussion about ULPL based methods, please refer to **Q0.4** in the general response.
>
> In terms of upstream debiasing, it is true that upstream debiasing is also unsupervised. However, there are two problems in incorporating these methods as additional baselines. First of all, to the best of our knowledge, upstream debiasing can only be applied to pretrained language models, which is not applicable to structured data (COMPAS and Adult in this paper). Secondly, upstream debiasing focuses on removing target bias, which is not comparable to the author protected attributes in the dataset. Taking the TrustPilot dataset as an example, the gender attribute is the gender of the review author. However, upstream debiasing aims at learning fair representations of reviews that are agnostic with the gender of an entity in the review. Clearly, there is a mismatch between reviewer gender in downstream debiasing and entity gender in upstream debiasing.
>
> Across the 3 NLP tasks, only the Bios-Gender dataset can be used for both upstream and downstream debiasing, if we assume everyone is the author of their own biography. However, Steed et al. (2022) recently showed over the bios dataset that upstream debiasing only minimally improves downstream fairness (Figure 2a in Steed et al., 2022), i.e., the results of upstream debiasing are almost identical to the Vanilla model in the downstream task.
>
> References:
>
> [1] Ryan Steed, Swetasudha Panda, Ari Kobren, and Michael Wick. 2022. Upstream Mitigation Is Not All You Need: Testing the Bias Transfer Hypothesis in Pre-Trained Language Models. In Proceedings of the 60th Annual Meeting of the Association for Computational Linguistics (Volume 1: Long Papers), pages 3524–3542, Dublin, Ireland. Association for Computational Linguistics.

---

> > ### Author Response · Authors · 2022-11-18
> > **Response to Reviewer y5Cm (2/2)**
> >
> > ## Q4.4 Related Work
> > We have replaced 'citet' with 'citep'; thanks!
> >
> > ## Q4.5 Computation of AUC-PFC.
> >
> > Please refer to the calculation of AUC-PFC in Appendix B. We respond to the specific question below.
> >
> > Assume we are looking at Figure 5 in the Appendix, where we present the PFC of in-domain and cross-domain debiasing over the Bios–gender dataset.
> > The AUC-PFC is normalized by the left- and right-most points as the review stated, and the maximum area that can be achieved by a perfect model is around $(0.8-0.3) \times 1 \approx 0.5$, where $0.8$ and $0.3$ are the best (most right) and worst (most left) performance, respectively. $\times 1$ refers to the ideal model achieves best performance ($0.8$) and best fairness ($1$) at the same time.
> >
> > ## Q4.6 Low AUC-PFC values for particular datasets.
> >
> > Please refer to the interpolation of AUC-PFC (**Q0.2**) in the general response. We respond to the specific question below.
> >
> > In particular, an extremely low AUC typically indicates that the best performance achieved is not much better than the worst performance (a result of normalization discussed in Q4.5). Taking the TrustPilot dataset as an example, the vanilla performance is around 0.82, the majority label proportion is 0.68, and the best AUC-PFC can be achieved $(0.82-0.68) \times 1 = 0.14$.
> >
> > In general, AUC-PFC presents the relative goodness of a method, but its absolute values have been normalized and are not directly comparable across datasets.
> > Moreover, for a vanilla model (which is a single point), assuming its performance and fairness are $\alpha$ and $\beta$, respectively, and the majority label proportion is $\gamma$, we can directly calculate its AUC-PFC score as
> > $$(\alpha-\gamma) \times 1 - \frac{1}{2} (1 - \beta) \times (\alpha-\gamma) = (\alpha-\gamma) \times (\frac{1}{2} + \frac{\beta}{2}),$$
> > where $(\alpha-\gamma)$ is typically the best AUC-PFC that can be achieved on this dataset.
> >
> > ## Q4.7 The selection of k for label smoothing without accessing ground truth fairness measures.
> >
> > Please refer to the discussion of the unsupervised hyperparameter tuning (**Q0.3**) in the general response.

---

> > > ### Comment · Reviewer_y5Cm · 2022-11-21
> > > **Thank you for your response**
> > >
> > > Thank you for your response! After reading through your answers, I think the paper could be more self-contained, e.g., the calculation of AUC-PFC should be contained in the main part of the paper, given that is important for your experiments and not quite straightforward as a measure. The same holds for other explanations you convey through your responses. In general, I am still concerned about the weak theoretical foundation of the work. I will keep my score.

---

### Official Review · Reviewer_djVz · 2022-10-22

**Confidence:** 4
**Correctness:** 3
**Technical Novelty And Significance:** 3
**Empirical Novelty And Significance:** 3
**Recommendation:** 8

**Clarity, Quality, Novelty And Reproducibility:**

- The paper is written clearly, and mostly with sufficient depth although some assumptions are glossed over. As the authors note, other approaches exist to unsupervised approaches in this space, but the simple approach proposed here is technically novel. The baselines are run with the fairlib library. The Appendix is fairly complete


**Strength And Weaknesses:**

Stengths

- Bias remains a crucial (and ongoing) challenge across many subdomains of ML. It is commendable that this work generalized beyond one of these subdomains in its analysis. Moreover, the focus on unlabeled demographics is more ecologically valid and therefore useful.
- Good, relevant, and (mostly) recent literature is effectively compared and contrasted in a well-structured manner.
- The datasets and baselines for empirical evaluation are well chosen
- The questions asked in each of the sub analyses of Sec 5 are well-chosen and mostly well executed

Weaknesses

- There are multiple group-wise measures of fairness beyond the one adopted in Sec 3.1. A minor point, but it would be preferable to show some rationale for the method chosen, and some indication of the alternatives.
- Certain assumptions seem to be glossed over in Sec 3.2, such as how (or whether) over- or under-represented groups can be binarized according to z_i’. This may of course be a useful heuristic empirically, but whether it has anything to do with actual groups or is merely correlated with them for other, more spurious reasons should be better explained (e.g.., in sec 5.2).
- Empirical results are fairly similar across rows, for multiple datasets and attributes, in Table 1. That this is the case for the unsupervised method is actually a positive outcome, but claims of ‘consistently outperforming’ other approaches seems to require some tempering, and some statistical tests of significance should nevertheless be run.


**Summary Of The Paper:**

Bias creeps into ML models trained from human behavioural data, which can usually be mitigated against but (typically) with labeled data for that purpose. This paper proposes a meta-algorithm for debiasing representations called Unsupervised Locality-based Proxy Label assignment. It is evaluated over five datasets, from NLP to structured data, and this evaluation shows that the provided technique recovers proxy labels that correlate with ‘unknown demographic data’. Performance is competitive with supervised methods.

**Summary Of The Review:**


- The positives of moderate novelty, fairly good empirical approaches (sub analyses, datasets, baselines), and the importance of this space outweigh negatives of glossed-over rationales and assumptions, and perhaps some overstatement of performance.

---

> ### Author Response · Authors · 2022-11-18
> **Response to Reviewer djVz**
>
> We thank Reviewer djVz for the detailed comments and thoughtful feedback on our work. Responses to specific comments are provided below.
>
> ## Q3.1 Alternative group-wise measures of fairness.
>
> Thanks for pointing out these related works. We have added discussion in the paper (Appendix B):
>
> Other than the absolute gap metric ($ |{U}\_{{\textnormal{c}}, {\textnormal{g}}}- {U}\_{{\textnormal{c}}}|=0 $), a broad range of formats of metrics have been introduced in previous studies to capture different assumptions about the nature of fairness. For example, Lum et al. (2022) propose to measure the variability of performance across demographic groups ( $\frac{1}{G-1}\sum_\{{\textnormal{g}}}|{U}\_{{\textnormal{c}}, {\textnormal{g}}}-{U}\_{{\textnormal{c}}}|^{2}$ ), Yang et al. (2020) only focus on the largest gap ($\text{max}\_{{\textnormal{g}}}|{U}\_{{\textnormal{c}}, {\textnormal{g}}}-{U}\_{{\textnormal{c}}}|$), and Feldman et al. (2015) measure performance ratio rather than gap ($\frac{\text{max}\_{{\textnormal{g}}}{U}\_{{\textnormal{c}}, {\textnormal{g}}}}{\text{min}\_{{\textnormal{g}}}{U}\_{{\textnormal{c}}, {\textnormal{g}}}}$).
> Although different aggregation methods have been applied to measure group-wise fairness, the optimization of any one of them is sufficient for the optimization of other methods, as the optimization conditions of these metrics are identical, ${U}\_{{\textnormal{c}}, {\textnormal{g}}} = {U}\_{{\textnormal{c}}} \forall {\textnormal{c}}, {\textnormal{g}}$.
>
> References
>
> [1] Kristian Lum, Yunfeng Zhang, and Amanda Bower. 2022. De-biasing “bias” measurement. In 2022 ACM Conference on Fairness, Accountability, and Transparency (FAccT '22).
>
> [2] Yang, Forest, Mouhamadou Cisse, and Sanmi Koyejo. "Fairness with overlapping groups; a probabilistic perspective." Advances in neural information processing systems 33 (2020): 4067-4078.
>
> [3] Feldman, Michael, et al. "Certifying and removing disparate impact." proceedings of the 21th ACM SIGKDD international conference on knowledge discovery and data mining. 2015.
>
> ## Q3.2 Theoretical justification.
> We discuss the theoretical justification in the general response (**Q0.1**), and respond to specific questions below.
>
> In the general response, we show that the proxy labels correlate better with ground truth protected labels for more biased models. Our experimental results in Table 2 and Section 5.1 also support this argument.
>
> Re the explanation of unfairness, although the proxy label assignment can be directly linked to performance disparities and provide a theoretical guarantee of empirical fairness, it does not explain the cause of the unfairness, such as identifying spurious correlations.
>
> ## Q3.3 Interpretation of empirical results.
>
> Please refer to the general response for the discussion of the interpretation of AUC-PFC experiments (**Q0.2**). We respond to the specific questions below.
>
> Re statistical tests of significance, it is still an open problem for the AUC-PFC metric. One problem is that each point on the PFC curve represents a particular model and list of predictions. Therefore, there is no paired sample case in the PFC plot to test significance.
>
> Instead, in Appendix E, we report descriptive statistics of each task, including mean $\pm$ stdev over five runs with different random seeds. In addition to results over the test dataset, we also report results over the development dataset for better transparency.
>
> It can be seen that (e.g., Table 14 in the Appendix for the Moji dataset) the standard deviations are  small compared to mean differences.

---

### Official Review · Reviewer_r7Nv · 2022-10-24

**Confidence:** 4
**Correctness:** 3
**Technical Novelty And Significance:** 2
**Empirical Novelty And Significance:** 3
**Recommendation:** 6

**Clarity, Quality, Novelty And Reproducibility:**

**Clarity:**

The paper was written clearly, and it is easy to follow.

**Quality:**

The paper is of high quality in terms of empirical experiments, but not so much theoretically.

**Novelty:**

While the problem on its own is not very novel, the approach proposed can be considered to some extend novel.

**Reproducibility:**

Authors provided an anonymous source code which will be really useful for reproducibility. Discussions in the Appendix also is comprehensive which can make it easier to reproduce the exact results in the paper.

**Strength And Weaknesses:**

**Strengths:**
1. The paper studies an important subject. Achieving fair outcomes in settings where access to protected attributes is limited is important and interesting.
2. Authors perform various experiments on datasets both from NLP and ML domains to showcase various aspects of the paper.
3. The method is easy and straightforward.
4. Paper is well-written and easy to follow.


**Weaknesses:**
1. It seems like the method is data dependent. Maybe results from section 5.1 on TrustPilot dataset is exactly due to this reason? I think further analysis or discussion need around this issue.
2. Related to my above comment, while the paper does a good job in terms of empirical analysis, theoretical discussion is lacking. This theoretical discussion maybe address my above comment (1). That this approach can be applicable regardless of specific data properties.
3. The performance of the approach is not impressive considering that comparison with the only unsupervised approach ARL is not done in a fair manner.
4. How much do you think the base supervised approach that ULPL is implemented over is having effect on the results. Although authors discuss this briefly in "Different ULPL methods:" section. More in depth discussion might be useful to improve the paper.
5. The approach is dependable on some parameters such as k which might make it inefficient. Is there a good way to make this more flexible?

**Additional Questions that needs clarification:**

The framework based on indicating poorly vs well modeled instances seems more aligned with equalized odds or equality of opportunity notions of fairness. Do authors know why their approach works for statistical parity which is different in nature than what is proposed here?


**Summary Of The Paper:**

The paper proposes a new framework to achieve fair outcomes in settings where access to sensitive attributes is not available. The method is called Unsupervised Locality-base Proxy Label assignment (ULPL) and is based on assigning proxy labels according to model predictions for poor vs well-modeled instances. The authors perform various experiments on datasets from ML and NLP domains.

**Summary Of The Review:**

Overall, the paper is a well-written and motivated paper; however, some deeper theoretical discussions can improve the paper drastically. The approach has also some flaws in terms of dependability to data, parameters, as well as some performance related concerns which I discussed in detail under weaknesses.

---

> ### Author Response · Authors · 2022-11-18
> **Response to Reviewer r7Nv**
>
> We thank Reviewer r7Nv for the detailed comments and thoughtful feedback on our work. Responses to specific comments are provided below.
>
> ## Q2.1 Data dependent and theoretical justification.
>
> We discuss theoretical justifications in the general response (**Q0.1**), and summarize our response to specific questions below.
>
> As discussed in Section 5.1 and pointed out by Reviewer y5Cm, because naively-trained models on the TrustPilot dataset are largely fair (achieving around 0.96 fairness in Table 2, and smaller leakage than other datasets in Table 3), the ground truth protected labels cannot be aligned with loss disparities. Therefore, there is a weak correlation between proxy labels and ground truth labels on this dataset. Such observations indicate that the more biased a model is, the better results can be achieved by our proposed method. And conversely, the technique should be used with caution if the dataset is already fair.
>
> ## Q2.2 Interpretation of experimental results.
>
> We discuss the magnitude of AUC-PFC in the general response (**Q0.2**) and summarize our response to specific questions below.
>
> There is little work on unsupervised bias mitigation, and ARL is the current SOTA method, which has been shown to substantially outperform previous work. In this paper, we show that our proposed method beats ARL on multiple tasks in terms of AUC-PFC scores.
> Although the improvements may appear minor in magnitude -- e.g. 0.016 AUC-PFC improvement on the Moji dataset -- a 0.001 improvement corresponds to a 3 percentage point (pp) $\times$ 3 pp area in the PFC plot, which is substantial. Such results are also supported by the disaggregated results in Appendix E.
>
> ## Q2.3 The application of ULPL to different debiasing approaches.
> Please refer to **Q0.4** in the general response.
>
> ## Q2.4 Tuning k for label smoothing.
>
> We acknowledge that tuning the $k$ value for label smoothing via grid search can be computationally expensive. We have indeed attempted to find a way of tuning $k$ without retraining the whole model, for example, conducting offline tuning of $k$ for a particular metric. Unfortunately, experiments (Table 3b) show that the optimal value is not only dataset-specific but also debiasing method specific. As such, the best $k$ value has to be tuned together with model training.
>
> Although we do not present an algorithm for searching $k$ effectively, we have observed that the optimal value of $k$ is positively correlated with model leakage and unfairness. Therefore, we could start tuning the value of $k$ from a large value if the model is significantly biased, as the instances from the same protected group are likely to be close to each other. Otherwise, we can use the proxy labels without smoothing if the results are reasonably fair.
>
>
> ## Q2.5 Optimization for Demographic Parity Fairness.
>
> We discuss the optimization for DP fairness in the general response (**Q0.1**), and summarize our response to specific questions below.
>
> DP fairness is achieved if an identical positive prediction rate (PPR, $p(\hat{{\textnormal{y}}}=1)$) is achieved across demographic groups. In the general response, we show that $p(\hat{{\textnormal{y}}}=1) = p(\hat{{\textnormal{y}}}=1, {\textnormal{y}}=0) + p(\hat{{\textnormal{y}}}=1, {\textnormal{y}}=1)$ can be approximately optimized w.r.t. $\mathcal{L}^{1}-\mathcal{L}^{0}$. Moreover, we show that proxy labels that are derived from $\mathcal{L}^{1}$ and $\mathcal{L}^{0}$ can be used to improve the lower bound of DP fairness.

---

> > ### Comment · Reviewer_r7Nv · 2022-12-03
> > **Response to Authors**
> >
> > Thank you for providing answers to my concerns. I appreciate the time and effort you put into responding to reviewers. I also agree with reviewer y5Cm and that you should put some of these discussions in the paper as they are important and are mostly the same concerns amongst different reviewers. I also think some experimental results on DP would be helpful.

---

### Official Review · Reviewer_sY6r · 2022-10-25

**Confidence:** 3
**Correctness:** 3
**Technical Novelty And Significance:** 2
**Empirical Novelty And Significance:** 2
**Recommendation:** 6

**Clarity, Quality, Novelty And Reproducibility:**

Clarity: while the paper is easy to follow, after reading I’m unsure of the following:
- For each dataset, are the protected attributes in question always binary? How many values can each take on? This could help shed more light on the results.
- For ULPL, is the loss per instance taken from the final model or summed over all training iterations?
- What is the precise way to apply this? Train a model, then label the examples, assign new labels with knn, then debias? Or does this whole process happen at each iteration?
- During neighborhood smoothing, are points relabeled with their smoothed label or do they retain their unsmoothed labeled? I imagine that an example’s smoothed label can be different from its unsmoothed label, leading it to vote differently in smoothing decisions for other examples.
- In the sensitivity to k-NN hyperparamters section, what is the variable p?
- In the same section: what is class-specific vs. ins-pecific smoothing?

Quality:
- ULPL seems like reasonable approach to debiasing without group information
- The experiments seem to be technically sound, however the results do not seem very significant.
- The writing is clear, but some of the details about the method and experimental methodology have been left out.

Novelty:
- The primary novelty is the idea to group examples into those that experience more/less than average loss (ULPL) as a proxy for protected groups when none are provided. As previously mentioned, while I don’t believe that examples with more loss will always (or even most of the time) correspond to specific protected groups, using existing debiasing methods to more evenly distribute the loss seems like a reasonable way to encourage parity across groups, when groups are unknown.
- The idea to use k-NN smoothing is not particularly novel and moreover, as previously mentioned, it seems problematic from a hyperparameter-tuning perspective. Moreover, it only helps half of the time.
- The problem of debiasing without group information is interesting, relevant, and has not received much attention.

Reproducibility:
- A link to the code used to run the experiments is provided, so I believe the results could be replicated. However, I have not tried to replicate them myself.

Other comments:
- Please bold the best results in the table to make them easier to read.
- Consider adding a citation to work on dataset cartography, which also groups points based on how difficult they are to predict correctly


**Strength And Weaknesses:**

Strengths:
- The crux of the approach in this paper is to try to equalize training loss among all examples. The proposed method, ULPL, simply groups the examples into examples that experience lower than average loss, and those that experience higher.  Once labeled, other debiasing methods are used to equalize the loss. This seems reasonable as a way to improve fairness. The argument made in the paper is that examples with lower loss are likely to be of the same (under-represented) demographic group. While there is some empirical evidence that supports this claim in previous work, I find it weaker and somewhat inconsequential: if training loss is more evenly divided across examples (and training loss is a good proxy for performance) then fairness should improve across all groups.
- This paper is clearly written and easy to follow.
- Experiments are performed with 5 datasets. Significant analysis is also presented.

Weaknesses:
- The proposed label smoothing is not consistently effective. Table 3 shows that in 50% of experimental conditions, no label smoothing is better than label smoothing.
- Label smoothing seems to be impossible to tune in realistic settings (in terms of the number of neighbors k). A standard approach of tuning k would be selecting k that yields the best results on the validation set. Since the metric being optimized is AUC-PFC, demographic group information is required to compute the metric. But the motivation of this paper includes the assumption that no demographic information is available.
- The results are difficult to interpret. Table 1, which includes the main results, reports AUC-PFC, which characterizes the tradeoff between performance and fairness. I assume that the proposed method helps to improve fairness, but does it also improve performance? A more detailed characterization of how ULPL affects performance and fairness would help. Moreover, the difference between the proposed approach and other competing methods is often a few thousandths of a unit of area, which seems small, but as a reader it is difficult to be certain.



**Summary Of The Paper:**

This paper focuses on the problem of debiasing representations learned by encoder-decoder models when demographic information is missing. The authors propose Unsupervised Locality-based Proxy Label assignment (ULPL), which assigns a binary “proxy” label to each example based on whether the model incurs a larger or smaller than average loss on that example. Once examples have been assigned proxy labels, it is possible to apply existing debiasing methods. In order to be robust to noisy outliers, the paper also proposes smoothing the proxy label for each example via a majority vote among that example's neighbors in the encoded space. In experiments on 5 datasets (with various protected attributes in focus for each dataset) they show that ULPL leads to larger areas under the performance fairness curve than ARL and a baseline that does not employ fairness interventions; ULPL is claimed to lead to similar performance as some supervised methods. Analysis showing that proxy labels generated by ULPL are generally correlated with protected labels is provided, as well as some analysis of kNN smoothing and an experiment where groups are identities by cross-products of protected attributes.


**Summary Of The Review:**

The paper is written clearly but some important details are missing. One of the methods proposed in this paper (label smoothing) does not consistently provide benefit and seems impossible to tune. The primary method proposed (ULPL) seems reasonable but doesn’t lead to significant gains.

---

> ### Author Response · Authors · 2022-11-18
> **Response to Reviewer sY6r (1/2)**
>
> We thank Reviewer sY6r for the detailed comments and thoughtful feedback on our work. Responses to specific comments are provided below.
>
> ## Q1.1 Effectiveness of label smoothing.
> Although label smoothing does not improve results for the two datasets in our experiments, we would like to emphasize that Bios and TrustPilot are representative of two different types of problem cases, and can be addressed in practice.
>
> Specifically, for TrustPilot, Tables 1 and 3a show that the naively-trained model has competitive fairness, and that there is little scope for improvement on this dataset. In our experience, such fair datasets are rare in real-world applications.
>
> Regarding the Bios dataset, label smoothing fails due to the skewed target label distributions, where the number of instances with the same target label is insufficient for smoothing within a minibatch. Unless there are multiple instances for each target class, this problem can be addressed by increasing the batch size. Unfortunately, we cannot provide further experimental results due to limitations in our computational infrastructure.
>
> Improving the robustness of label smoothing is an interesting direction for future work.
>
> ## Q1.2 Hyperparameter tuning for the label smoothing without observing demographic information.
> Please refer to **Q0.3** in the general response.
>
>
> ## Q1.3 Interpretation of AUC-PFC results.
> We discuss the interpretation of AUC-PFC experiments in the general response (**Q0.2**) and summarize the response to the specific questions below.
>
> Re the magnitude of AUC-PFC results, $0.0001$ is equivalent to a 1 percentage point (pp) improvement in both performance and fairness in the PFC plots (e.g. Figure 2b), whereas the smallest improvement in the paper is $0.001$, corresponding to a 3pp $\times$ 3pp improvement.
>
> Similarly to the supervised methods, ULPL-based methods trade off between performance and fairness, i.e., a debiasing method improves fairness at the cost of performance. Except for Moji, where the test set is balanced, improving fairness results in lower performance. Appendix E provides PFC plots for each setting, including a quantitative comparison across debiasing methods. Observations are consistent with AUC-PFC results, where our proposed unsupervised methods are competitive with supervised debiasing baselines.
>
> ## Q1.4 Details of protected attributes.
> We provide key characteristics of all datasets in Appendix A, including the details of protected attributes. Most single protected attributes are binary, except the ethnicity attribute for COMPAS and Adult. In particular, there are 3 race groups in
> COMPAS: (1) African-American, (2) Caucasian, and (3) Other; and 5 race groups in Adult: (1) White, (2) AsianPac-Islander, (3) Amer-Indian-Eskimo, (4) Other, and (5) Black.
>
> Moreover, we also consider debiasing for intersectional groups in Section 5.3, which can be treated as a single multi-class protected attribute. Taking the Bios dataset as an example, the 4 intersection groups w.r.t. gender and economy level are: (1) male-high, (2) male-low, (3) female-high, and (4) female-low.
>
> Our experiments (Tables 1 and 4) show that the proposed method generalizes to both single protected attribute with multi-class values, and intersectional groups.
>
> ## Q1.5 The process of proxy label assignment.
> We make the following clarification in Appendix C.1.
>
> In the current experiments, proxy labels are assigned based on the losses of each minibatch, i.e., the loss per instance is taken from a particular training iteration. We acknowledge that there are other ways of extracting training losses, e.g. taking losses from the final model or averaging over multiple iterations as you suggest, which we leave to future work.

---

> > ### Author Response · Authors · 2022-11-18
> > **Response to Reviewer sY6r (2/2)**
> >
> > ## Q1.6 The process of label smoothing.
> >
> > We make the following clarification in Appendix C.1.
> >
> > The proxy label assignment and smoothing happen simultaneously at each iteration. In doing so, our method can be incorporated into existing supervised debiasing systems with only a few lines of code to replace gold protected labels with our proxy labels.
> >
> > In the [anonymous source code](https://anonymous.4open.science/r/An-Unsupervised-Locality-based-Method-for-Bias-Mitigation-2B30/README.md), step 3 introduces the usage of proxy labels, where at each minibatch, the actual protected labels are replaced with smoothed proxy labels. All debiasing methods are then applied to the proxy labels in latter processing.
> >
> > During label smoothing, unsmoothed labels are used for voting to avoid inconsistency in smoothing decisions for other examples. In practice, as shown in lines 65-76 of the [the anonymous code](https://anonymous.4open.science/r/An-Unsupervised-Locality-based-Method-for-Bias-Mitigation-2B30/fairlib/src/networks/knn_labels.py), we first collect the nearest neighbours of each instance and then perform voting.
> >
> > ## Q1.7 Explored hyperparameters in Figure 3.
> >
> > When searching the neighbours of an instance, we use the $p$-norm of differences between two instances to measure their distance, where $p$ is the power of norm. For example, $p=1$ and $p=2$ correspond to the taxicab norm and Euclidean norm, respectively. Our experiments show that label smoothing is robust to the value of $p$, and as a result, one can avoid tuning $p$ in experiments.
> >
> > As for class-specific vs. -inspecific, we explore if considering a subset of instances reduces the performance of the KNN mechanism. In particular, the nearest neighbours of an instance are required to have the same target label (Section 3.2), such that their proxy labels are derived with the same threshold. However, there is a risk of reducing the number of candidate instances for label smoothing. To analyze this problem, we examine the KNN performance using instances with the same target class (class-specific), and using all instances (-inspecific). Figure 3 shows that using a class-specific approach ensures consistency both in proxy label assignment and KNN performance.
> >
> > ## Q1.8 Bold results and a citation to dataset cartography.
> >
> > Thanks for the suggestion. We have updated the tables as suggested, and  provide additional discussion regarding the cartography paper in the Related Work section (Section 2), replicated below:
> >
> > **Dataset cartography** Training instances are also grouped based on their predictability in the literature of dataset cartography, which is similar to the assignment of proxy labels in this paper. Swayamdipta et al. (2020) propose to visualize the diagnose training instances according to variability and confidence, where a higher-confidence indicates the instance is easier to be correctly predicted. Le Bras et al. (2020) also group training instances by their predictability, which is measured by training simple linear discriminators. Such methods focus on improving in- and out-of-distribution performance without taking fairness into consideration. On the contrary, our proposed method aims at mitigating bias by assigning proxy protected group labels to training instances based on their losses within a particular class.
> >
> > **References**
> >
> > [1] Swabha Swayamdipta, Roy Schwartz, Nicholas Lourie, Yizhong Wang, Hannaneh Hajishirzi, Noah A. Smith, and Yejin Choi. 2020. Dataset Cartography: Mapping and Diagnosing Datasets with Training Dynamics. In Proceedings of the 2020 Conference on Empirical Methods in Natural Language Processing (EMNLP), pages 9275–9293, Online. Association for Computational Linguistics.
> >
> > [2] Ronan LeBras, Swabha Swayamdipta, Chandra Bhagavatula, Rowan Zellers, Matthew E. Peters, Ashish Sabharwal, and Yejin Choi. 2020. Adversarial filters of dataset biases. In ICML.

---

> > > ### Comment · Reviewer_sY6r · 2022-12-12
> > > **Thank you for your detailed responses**
> > >
> > > Thank you very much for the thoughtful response to all my questions. Your responses have given me a better understanding of the work and, with the revisions you've made, I think the paper has improved. I'm still weary of the hyperparameter tuning, but your explanation of the results have given me a better sense of their significance. I will increase my score by 1.

---

### Author Response · Authors · 2022-11-18
**Response to all reviewers (1/2)**


We thank the reviewers for their thoughtful and constructive review of our manuscript.
In response to your feedback, we provide a general response here to points raised by multiple reviewers, followed by individual responses to specific reviewer concerns, and an updated manuscript.

## Q0.1 Theoretical Justification.
In response to questions from Reviewers sY6r, djVz, and y5Cm about a theoretical justification for the proposed method, we have added a discussion in Appendix F.

In particular, we take equal opportunity fairness and demographic parity fairness as examples and show that minimizing loss disparity can be a practical approximation of empirical fairness (Appendix F.1). In Appendix F.2, we show that proxy labels correlate better with ground truth group labels for more biased models, and optimization w.r.t. proxy labels improves the lower bound of fairness. Moreover, we discuss why proxy labels can be used for optimizing different fairness criteria.

## Q0.2 Interpretation of experimental results.
Regarding comments from all Reviewers about interpreting AUC-PFC scores, we added the following clarifications in Appendix B.1.

The main motivation for using AUC-PFC is ease of comparison between approaches, as it aggregates the performance-fairness trade-off curve (PFC) of each model to a single number, enabling systematic comparison across different tasks. The two common questions related to AUC-PFC are:

- The *magnitude of AUC-PFC* differs from a single metric, and a $0.0001$ improvement in the AUC-PFC score is equivalent to a 1 percentage point (pp) better in both performance and fairness ($0.01 \times 0.01$). In the paper, numbers are rounded to 3 decimals, and a minimum difference in AUC-PFC ($0.001$) is equivalent to approximately 3 pp improvement in both performance and fairness in a PFC plot.
- The *calculation of AUC-PFC* scores are normalized by the worst performance, which is the majority label proportion when using the accuracy metric (See Appendix B for details). Therefore, AUC-PFC scores represent to what extent a model improves the performance or fairness over the random model.

There is no doubt that using AUC-PFC comes with certain limitations, and to address concerns related to AUC-PFC scores, we present additional results in Appendix E, including dis-aggregated results of each dataset.
In particular, we provide the *PFC* of each method (e.g., Figure 6 in Appendix E), representing the best fairness that can be achieved at different performance levels and vice versa.

One limitation of a PFC plot is that it is hard to make quantitative conclusions based on the plot itself, and we cannot conclude that one method is better than another if any intersection exists between their PFCs.
To address this problem, we additionally conduct *quantitative comparisons* across different debiasing methods by model selection w.r.t. 2 different criteria and then compare both the performance and fairness of the selected models (e.g., Table 14 in Appendix E). For each method, we report the evaluation results averaged over 5 random runs with standard deviation for both the development set and test set.

As stated in Appendix E, we present disaggregated results (including a PFC plot and a table) for all 15 settings on an [anonymous website](https://anonymous.4open.science/r/An-Unsupervised-Locality-based-Method-for-Bias-Mitigation-2B30/unsupervised_bias_mitigation/NB_Appendix_indomain_tradeoffs_dispaly.ipynb).

## Q0.3 Hyperparameter tuning without protected labels.

We agree with Reviewers sY6r and y5Cm that hyperparameter tuning for fairness typically requires protected attributes in the validation set, which has been stated explicitly in the paper (Section 4.1 - Model comparison). To justify applications without demographics, we have made the following clarifications in appendix C.5.

We would like to point out that hyperparameter tuning without protected attributes is similar to evaluation without ground truth labels, in the sense that instances are not annotated with the labels  required for evaluation. One line of related (but orthogonal) work in this direction is the use of proxy variables, such as
Kallus et al., (2021) measuring fairness w.r.t. surname and geolocation across different datasets. In practice, one can employ our unsupervised debiasing methods together with unsupervised fairness evaluation approaches to perform hyperparameter tuning for better fairness.

References

[1] Nathan Kallus, Xiaojie Mao, Angela Zhou (2021) Assessing Algorithmic Fairness with Unobserved Protected Class Using Data Combination. Management Science 68(3):1959-1981.

---

> ### Author Response · Authors · 2022-11-18
> **Response to all reviewers (2/2)**
>
> ## Q0.4 The application of ULPL to different debiasing approaches.
> In response to questions from Reviewers r7Nv and y5Cm, We have added the following discussion to Appendix D.6.
>
> $GD_{\mathrm{CLA}}$ is representative of debiasing methods that directly optimize loss parity.
> In particular, the training objective of $GD_{\mathrm{CLA}}$ is:
> $$\mathcal{L}\_\mathrm{GD_{CLA}} = \mathcal{L} + \lambda_\mathrm{GD_{CLA}}\sum_{{\textnormal{c}}}\sum_{{\textnormal{g}}}|\mathcal{L}^{{\textnormal{c}},{\textnormal{g}}}-\mathcal{L}^{{\textnormal{c}}}|,$$
> where $|\mathcal{L}^{{\textnormal{c}},{\textnormal{g}}}-\mathcal{L}^{{\textnormal{c}}}|$ is optimized to achieve better fairness. Since $\mathcal{L}^{{\textnormal{c}},{\textnormal{g}}}$, $\mathcal{L}^{{\textnormal{c}}}$ and $\mathcal{L}$ are average losses, their magnitude is normalised by subset sizes ($n_{{\textnormal{c}},{\textnormal{g}}}$, $n_{{\textnormal{c}}}$ and $n$, respectively), which in turn applies the same strength of fairness regularization to all subsets of instances ${\mathcal{D}}\_{{\textnormal{c}},{\textnormal{g}}}, \forall {\textnormal{c}},{\textnormal{g}}$. In other words, $GD_{\mathrm{CLA}}$ ignores the influence of group size in bias mitigation, resulting in robustness to imbalanced class distributions.
>
> There is a perfect alignment between ULPL and  $GD_{\mathrm{CLA}}$ in the sense that the proxy group ${\textnormal{z}}'=1$ will always be upweighted during the optimization of $GD_{\mathrm{CLA}}$. I.e., ULPL+$GD_{\mathrm{CLA}}$ reduces the loss disparities across instances within each target class, which in turn improves the lower bound of group-wise fairness, especially for EO fairness (see Appendix F.2 for details). As a result of this consistency, we observe that ULPL+$GD_{\mathrm{CLA}}$ outperforms other unsupervised methods.
>
> $GD_{\mathrm{GLB}}$ is a variant of $GD_{\mathrm{CLA}}$, and can only be differentiated by way of incorporating fairness regularization:
> $$\mathcal{L}\_{\mathrm{GD_{GLB}}} = \mathcal{L} + \lambda_{\mathrm{GD_{GLB}}}\sum_{{\textnormal{c}}}\sum_{{\textnormal{g}}}|\mathcal{L}^{{\textnormal{c}},{\textnormal{g}}}-\mathcal{L}|,$$
> where the average loss in the regularization term is based on all instances ($\mathcal{L}$), differing from the average loss within each target class ($\mathcal{L}^{{\textnormal{c}}}$) for $GD_{\mathrm{CLA}}$. As a result, $GD_{\mathrm{GLB}}$ additionally encourages the performance parity across target classes, typically known as long-tail learning.
> However, ULPL+$GD_{\mathrm{GLB}}$ could potentially lead to worse results for better-performing target classes. For example, for the $i$th target class, assuming that loss differences have been minimized (i.e., $\mathcal{L}^{i,{\textnormal{z}}'=0} \approx \mathcal{L}^{i,{\textnormal{z}}'=1}$), it is possible that all instances with target label ${\textnormal{y}}=i$ will be under-fitted if $\mathcal{L}^{i} > \mathcal{L}$.
>
> Adv represents a different family of debiasing methods that aims to learn fair hidden representations.
> The training objective of Adv includes the mutual information ($MI$) over the training objective in addition to standard loss:
> $$\mathcal{L}\_{\mathrm{Adv}} = \mathcal{L} + MI({\textnormal{z}}, {\pmb{h}}),$$
> where the ${\pmb{h}} = e({\pmb{x}})$ is the hidden representation of input ${\pmb{x}}$ extracted from the encoder $e$.
> By minimizing the $MI$ objective, we expect the learned hidden representations ${\pmb{h}}$ to be orthogonal to protected attributes ${\textnormal{z}}$.
>
> In practice, $MI({\textnormal{z}}, {\pmb{h}})$ is expressed by the combination of marginal entropy ($H({\textnormal{z}})$) and conditional entropy ($H({\textnormal{z}}|{\pmb{h}})$): $MI({\textnormal{z}}, {\pmb{h}}) = H({\textnormal{z}}) - H({\textnormal{z}}|{\pmb{h}})$, where $H({\textnormal{z}})$ is a constant number and can be ignored in the optimization, and $H({\textnormal{z}}|{\pmb{h}})$ is estimated by an adversary ($d$) that is trained to identify the protected attributes ($\hat{{\textnormal{z}}} = d ({\pmb{h}})$).
>
> The key step of Adv is the training of $d$, i.e., if $d$ can effectively recover ${\textnormal{z}}$ from ${\pmb{h}}$. One problem associated with ULPL is that the mapping from proxy labels to ground truth labels is class specific, making it harder to recover. Therefore, although the adversaries are non-linear classifiers, the effectiveness of ULPL+Adv is not as good as other ULPL$+*$ methods as shown in Table 1.

---

### Author Response · Authors · 2022-12-03
**Moving discussions from Appendix to the Paper**

Dear reviewers y5Cm and r7Nv, thank you for following up! We are delighted to see that our responses mostly address your concerns. We will move the discussions corresponding to general questions from the appendix to the main body upon acceptance (because we can not submit further revisions at this point).

---

### Decision · Program_Chairs · 2023-01-20

**Decision:**

Accept: poster

**Justification For Why Not Higher Score:**

Although the authors provide additional discussion in the rebuttal period, the theoretical justification of the paper is relatively weak as pointed out by multiple reviewers.


**Justification For Why Not Lower Score:**

All reviewers are positive about this paper, and there is no major concern.

**Metareview: Summary, Strengths And Weaknesses:**

The paper presents a framework to mitigate bias without the need to access sensitive attributes. Overall, all the reviewers are positive about this paper and major concerns are resolved during the rebuttal.

Strengths:
- The setting is practical as in real-world scenarios, sensitive attributes are often not available or are prohibited to keep due to privacy concerns. Achieving fair outcomes under this condition is a challenging and important topic.
- The experiments are comprehensive and conducted studies in multiple domains.
- Overall, the paper is well-written and easy to follow.

Weaknesses:
- The theoretical justification of the proposed approach is relatively weak.
- There are some over-claiming and handwaving assumptions. A careful revision is needed before publishing.


Missing reference:
The setting of evaluating NLP fairness without giving sensitive attributes is discussed in the following paper.
LOGAN: Local Group Bias Detection by Clustering, EMNLP 2020


**Note From Pc:**

if the above contains the word "oral" or "spotlight" please see: "oral" presentation means -> notable-top-5% and "spotlight" means -> notable-top-25%. As stated in our emails, we are disassociating presentation type from AC recommendations

**Summary Of Ac-Reviewer Meeting:**

N/A